# A laboratory perspective on accelerating preparatory processes before earthquakes and implications for foreshock detectability

Thomas H. W. Goebel [1] ✉, Valerian Schuster[2], Grzegorz Kwiatek [2], Kiran Pandey[1] & Georg Dresen [2]

Dynamic failure in the laboratory is commonly preceded by many foreshocks which accompany premonitory aseismic slip. Aseismic slip is also thought to govern earthquake nucleation in nature, yet, foreshocks are rare. Here, we examine how heterogeneity due to different roughness, damage and pore pressures affects premonitory slip and acoustic emission characteristics. High fluid pressures increase stiffness and reduce heterogeneity which promotes more rapid slip acceleration and shorter precursory periods, similar to the effect of low geometric heterogeneity on smooth faults. The associated acoustic emission activity in low-heterogeneity samples becomes increasingly dominated by earthquake-like double-couple focal mechanisms. The similarity of fluid pressure increase and roughness reduction suggests that increased stress and geometric homogeneity may substantially shorten the duration of foreshock activity. Gradual fault activation and extended foreshock activity is more likely observable on immature faults at shallow depth.

Stick-slip motion on rough rock surfaces has long been recognized as a laboratory analog for earthquakes due to its abrupt acceleration toward seismogenic slip[1]. Stick-slip is governed by the competition between the rate of elastic unloading and frictional strength reduction over a critical slip distance[2]. An imbalance between these rates e.g., due to low stiffness of the loading system gives rise to rupture acceleration and earthquakes[3]. In theory such accelerating ruptures may generate precursory signals; however, in practice, rupture nucleation is rarely associated with detectable signals in nature[2,4,5].

Strain release across crustal faults is accommodated by a range of processes, including stick-slip, fault creep, and slow and fast slip transients[6–11]. Recent experiments demonstrated that, in the laboratory, dynamic rupture and slow slip events are commonly preceded by extended preparatory processes leading to a decrease in seismic velocities and b-values and an increase in acoustic emission (AE) rates before failure[12–14]. Note that in the following, we use 'failure' as a general term related to samples loosing their ability to support the applied

load during fracture and fault slip. The duration and detectability of precursory signals vary significantly with normal stress and fault roughness[15–17]. Precursory signals have long been recorded prior to intact rock fracture and the brittle failure of large asperities[3,18–21]. However, preparatory signals during stick-slip are more challenging to detect especially on smooth faults[15].

Laboratory observations indicate that long and short-term preparatory phases may lead to different precursory signals[22]. Long-term signals include increasing seismicity rates, decreasing seismic velocity, lower b-values and spatial seismicity localization[12–15,23]. Such long-term preparatory processes are characterized by damage accumulation, crack alignment with respect to each other and strain localization[15,19,24,25] which has also been observed in nature[26]. Short-term signals stem from processes associated with rupture nucleation and the commencement of detectable fault slip[4,27]. Long-term microdamage accumulation and spatial localization precedes measurable premonitory slip in many cases suggesting distinct processes at early

[1]University of Memphis, Center for Earthquake Research and Information, Memphis, TN, USA. [2]German Research Centre for Geosciences (GFZ), Section 4.2 Geomechanics and Scientific Drilling, Potsdam, Germany. ✉e-mail: thgoebel@memphis.edu

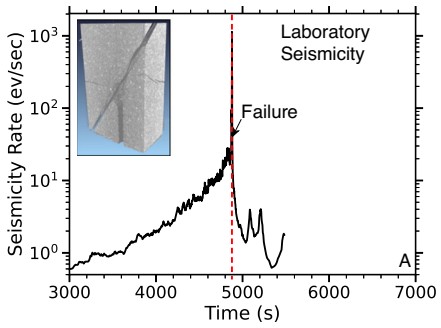

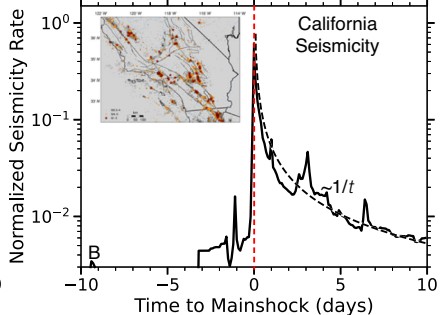

**Fig. 1 | Laboratory failure of faulted or intact samples is commonly preceded by many precursory signals such as abundant foreshocks, such signals are exceedingly rare in nature. A** Exemplary microseismicity rate increase (black curve) before rock failure (red dashed line). Inset shows CT-scan of postmortem sample that contained a central borehole to control fluid pressure. **B** Stacked seismicity rates before and after all M5 to M6.5 mainshocks (see Method) in southern California for 30 years between 1981 and 2011. Inset shows seismicity map of Southern California with mainshocks highlighted by red stars (map was created with the matplotlib-basemap library for Python). Note the differences in time scales in **A** and **B**.

and later pre-failure phases[12,24,26]. Nevertheless, machine-learning predictions of time-to-failure and fault stress state can be used without the need to understand underlying physics[28–32].

Many observations suggest a highly non-linear preparatory phase before laboratory failure. This non-linearity is rooted in the constitutive behavior that governs crack propagation, coalescence and subcritical crack growth[19,33]. Average propagation velocity, $V$, across a population of cracks increases as a power-law with higher stress intensity, $K_\sigma$[33]:

$$V = V_0 \left( \frac{K_\sigma}{K_c} \right)^n, \tag{1}$$

where $V_0$ is a reference velocity, $K_c$ is fracture toughness and $n$ is the stress corrosion exponent. Stress corrosion which is thought to facilitate subcritical crack growth is governed by the availability of fluids and may control AE-foreshocks in some lab-tests[18,34]. This framework can be extended to the non-linear behavior of interacting frictional contacts (see Discussion).

Homogeneity in stress field and fault geometry enhance the non-linearity of nucleation and preparatory processes before failure whereas heterogeneity promotes early rupture arrest and slow slip events[18,35–37]. For instance, fault roughness and heterogeneity promote a higher proportion of small magnitude AE events and smaller stress release during macroscopic slip[38–40]. Such heterogeneity may affect the propagation of individual cracks or entire crack populations, leading to substantial differences in magnitude-frequency distributions[41,42].

In addition to geometric and stress heterogeneity, rupture nucleation and propagation is also significantly affected by fluid pressures. Fluid pressure governs dynamic weakening processes on rapidly slipping faults, leading to large, secondary drops in frictional resistance due to thermal pressurization[43–45]. Fluid pressures may also reduce the rate of strength reduction with slip and the corresponding frictional parameters[2,9,46,47].

Laboratory fracture and stick-slip exhibit many parallels with natural faulting processes, leading to the same statistical distribution functions (i.e., Omori, Gutenberg-Richter and spatial decay)[40,48] and fault structure[21,49]. These similarities suggest that the same fundamental physics of frictional break-down and preparatory processes before slip govern labquakes and earthquakes[2,4,50]. Nevertheless, there are obvious differences in seismicity characteristics between lab and nature. Natural seismicity is strongly dominated by aftershock sequences with little to no foreshocks before most events (Fig. 1)[5,51]. Lab seismicity, on the other hand, is dominated by foreshocks with an exponential or power-law increase toward failure (see below for details).

The relative sparsity of foreshocks in nature may be due to differences in detection thresholds, source mechanisms, nucleation processes or crustal conditions at seismogenic depth.

Here, we show how different fault conditions, namely geometric complexity (i.e., fault roughness and damage) and pore fluid pressure affect precursory signals before failure. First, we demonstrate that AE-derived moment tensors become increasingly more dominated by earthquake-like double-couple components at high pore pressures. We then demonstrate structural differences in stable sliding vs. stick-slip faults which exhibit higher degrees of strain localization. Next, we show that elevated pore fluid pressures and reduced roughness lead to more rapid acceleration of premonitory slip. Associated precursory signals, namely pore space dilation, seismic velocity decrease, AE event localization and rate increase, are less pronounced and shorter on smooth faults and rough faults with high fluid pressures. This difference is likely caused by more homogeneous stress distributions and lower geometric complexity on both smooth and high fluid pressure faults.

## Results

We report results from ten experiments on cylindrical (50 × 105 mm), faulted Westerly granite samples at dry and fluid-saturated conditions with pore pressures of 0.5–35 MPa, confining pressures of 120–150 MPa and axial loading rates of 0.3 μm/sec (Supplementary Table S1). All fluid-saturated tests were conducted at constant-pressure boundary conditions to simulate fault patches that are embedded in a larger hydraulically connected damage zone (see Method). To generate naturally rough faults, eight samples were fractured along a pre-cut 30° notch at $P_c = 75$ MPa before loading the newly-created fault at $P_c = 150$ MPa until failure. This procedure led to stick-slip in six tests and stable sliding in two experiments as discussed in the following section. Prior to sample fracture, we saturated the samples for more than one hour by injecting fluid through a central borehole, leading to a systematic increase in P-wave velocity until pore pressures equilibrated (Supplementary Fig. S1). We created two smooth faults by cutting the rock cylinders at a 30° angle to the loading axis and subsequently sanding the surfaces to create planar interfaces.

We simulated the effect of elevated fault damage by heating six samples to 700 °C leading to pervasive microcrack damage caused by different thermal expansion of the various granitic mineral phases. These samples are referred to as high-damage experiments in the following. The resulting microcrack damage caused strongly non-linear stress-strain relationships at low confining pressures (Supplementary Fig. S2a). Loading at higher confining pressures resulted in more-linear stress-strain relationships but with relatively lower bulk moduli for thermally treated samples (Supplementary Figs. S2b, S8).

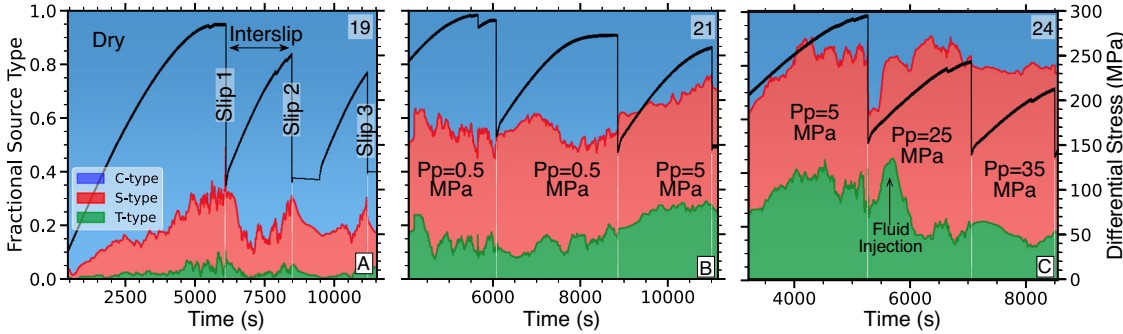

**Fig. 2 | Higher fluid pressure lead to more shear type events and fewer compaction events during stick-slip sliding on rough faults.** Proportion of AE source types, derived from AE polarity analysis, is shown in blue (compression), red (shear) and green (tensile, see Method) and respective differential stresses in black (Source data file provided). Note that relative contributions always add up to 100%. **A** Dry test (note that loading was paused for ~15 min between slip two and three.). **B** Fluid saturated with pore pressure, $P_p = 0.5$–5 MPa, **C** Fluid saturated with $P_p = 5$–3 MPa.

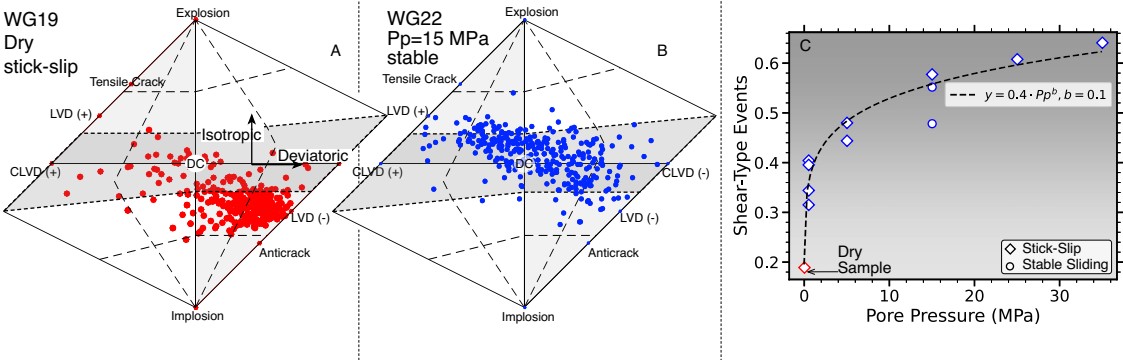

**Fig. 3 | AE moment tensor decomposition shows a significant shift from dominant isotropic to dominant deviatoric components with higher fluid pressures. A** Hudson plot of moment tensor components prior to dynamic failure on a dry fault. Gray shaded area highlights area of deviatoric components and pure double couple sources are at the center, labeled by 'DC'. **B** Same as **A** but for frictional sliding at $P_p = 15$ MPa. **C** Average fraction of shear type events as a function of pore pressure (Source data file provided).

## Mechanics of micro and macro-scale slip leading up to failure

Changes in pores pressures affect slip mechanics at the scale of sub-millimeter AEs and decimeter-scale slip events. We observe a progressive reduction in macro fault strength (i.e., peak differential stress before slip) with increasing pore pressures and successive stick-slip events (black curves in Fig. 2). This strength reduction is caused by the combined effects of lower effective stresses, surface smoothing and increased gouge production with cumulative slip.

Higher fluid pressures promote relatively more shear deformation expressed in dominant shear-type AE events at the micro-scale. The type of micromechanical deformation can be determined from AE first motion polarities (see Method) which show progressively less compaction in favor of shear type events with increasing pore pressure. Dry-fault compaction due to increasing axial loads is accommodated by seismically detectable pore space collapse and shear-enhanced compaction (Fig. 2A). The overall compaction of dry faults is punctuated by episodes of increasing shear-type events leading up to dynamic slip.

Fluid-saturated faults, overall, exhibit substantially more shear type AE events with increasing pore pressures while compressive sources are reduced (Fig. 2B, C). In addition, there is a larger fraction of tensile AE events, especially during periods of pore pressure increase when fluid is actively forced into the fault and the surrounding damage zone.

Taking advantage of the good focal sphere coverage in the lab (Supplementary Fig. S5), we perform full AE moment tensors inversions that include both isotropic and deviatoric components (see Method). The results confirm a dominance of negative isotropic sources (i.e., compaction) for dry samples, whereas tests at higher fluid pressures

shift toward dominant deviatoric sources (Fig. 3). Similarly, we observe a systematic increase in the relative percentage of shear-type events, $S_{\text{type}}$, with increasing pore pressure which is captured by a power-law relationship of the form $S_{\text{type}} \propto P_p^b$ (Fig. 3C). Shear type events increase systematically from less than 20% in dry samples to more than 60% at pore pressures of 35 MPa, with one notable outlier at $P_p = 15$ MPa.

The effects of increased pore pressures are not limited to the micro-mechanics of AEs but also lead to notable differences in macro strain-accumulation and release. The first slip event on rough faults at $P_p = 15$ MPa was stable for two of three tests at these pressures, whereas slip at lower pore pressure always produced stick-slip events (Supplementary Fig. S3). Stable sliding episodes at high pore pressure can be explained by effective stress reduction and increase in system stiffness (Supplementary Fig. S4). This increase in stiffness for fluid saturated samples is a result of the stiffening effect of fluid-filled vs. dry cracks[52]. Nevertheless, pore pressure alone does not uniquely determine whether fault slip is stable or unstable. For instance, fault slip remains frictionally unstable for slip at $P_p = 25$ MPa and 35 MPa if the same fault has already experienced stick-slip at lower pressures (Fig. 2C). Thus, the evolution of fault structure, gouge fabric and asperity distributions due to accumulated slip also affect frictional properties and slip stability[49].

## Comparison of fault micro-structural differences

We analyze fault microstructures in X-ray computer tomography scans and optical and scanning electron microscopy images (see Method). We find that internal fault structures consist of various structural

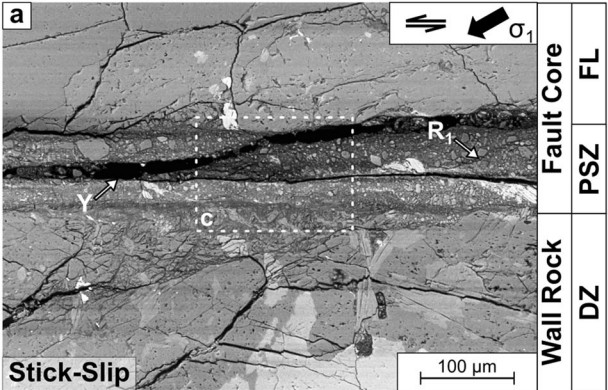

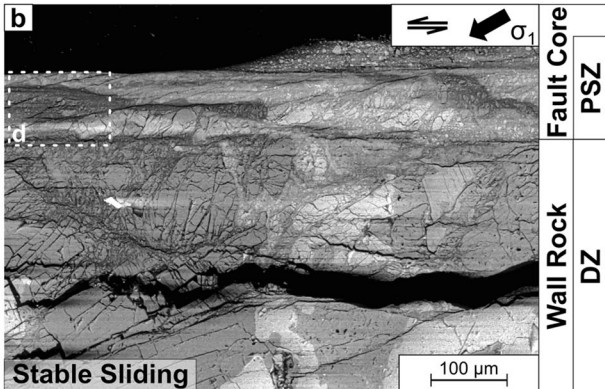

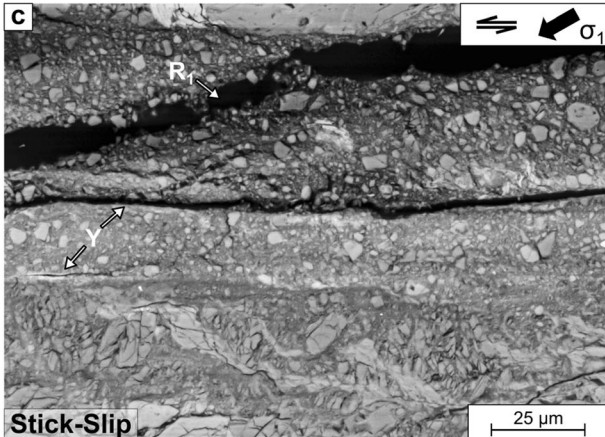

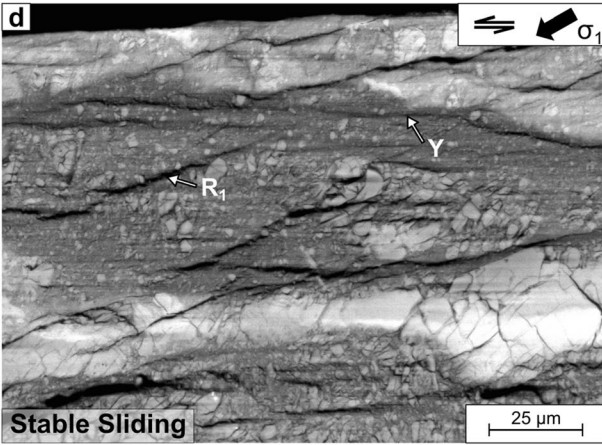

**Fig. 4 | Stable sliding (right panels) is associated with thicker, sheared gouge layers than stick-slip (left panels).** SEM images of representative principal slip zone structures with surrounding damage zone of stick-slip, (**a**, **c**: exp. 20) and stable sliding, (**b**, **d**: exp. 22). Panels **c** and **d** show the development of several Y and Riedel (R1) shears. DZ damage zone, PSZ principal slip zone, FL fractured lens (See online supplement for details).

elements comprised of damage zone, fault core and multiple, localized slip surfaces (Fig. 4). Gouge layers within the fault core are characterized by intense cataclasis and microcracking and display a high degree of fragmentation and grain size reduction, indicating extreme shear-strain localization. Principal slip zones, embedded in the gouge layers, consist of ultra fine-grained (<0.1−30 μm) material with variable thicknesses. High-damage samples due to inherited and expanding damage from thermal treatment exhibit pervasive microcracks throughout the sample (Supplementary Fig. S7) and up to 1.6 times higher crack density within the fault damage zone (Supplementary Fig. S8).

Differences in fault slip behavior lead to different post-mortem fault micro-structure. Stick-slip generated distinct, large offsets of up to 500 μm, indicating strain-localization on planarly-connected principal slip zones with <150-μm thickness (Fig. 4a). The grain size distribution of the densely compacted gouge layers is relatively narrow with a sharp transition to the adjacent wall rock (Fig. 4a). In contrast, the thicker gouge layers of stable sliding faults show a wider transitional zone to the wall rock, consisting of highly fractured and comminuted mineral grains. These grains are preferentially rotated towards the slip surface, supplying the gouge with wall rock material (Fig. 4b). The up to 1-mm thick gouge layer in these samples displays a wider grain size distribution ranging from sub-micron particles to largest survivor clasts of 100 μm. The thicker gouge and transition zones indicate that stable slip is less localized and occurs distributed across the core deformation zone.

Microstructural observations strongly support the observed differences in shear strain and damage creation in mechanical and AE data. The key observations here are that higher pore pressures favor shear displacements and stable sliding. Shear-dominated fault zones exhibited thicker gouge layers with less evidence of extreme strain localization. Thermally-induced microcrack damage leads to higher crack densities across the fault damage zone. In the following, we investigate how differences in damage and pore pressure affect preparatory processes before fault slip.

## Pore volume increase before slip on samples with high and low damage

We examine the effect of elevated fault damage by heating a subset of samples to 700 °C and find significant differences in pore volume evolution during stick-slip and rock fracture. Dynamic fracture of initially intact rocks is generally preceded by a pronounced dilatational phase, starting several thousand seconds before failure (Fig. 5A). The respective pore volume increases by ~0.35 cm$^3$ in high-damage samples and only ~0.04 cm$^3$ in low damage samples that were not thermally treated.

Stick-slip failure is preceded by similar dilational signals, however with much smaller amplitudes. Stick-slip on high damage faults at $P_p = 5$ MPa is preceded by pore volume dilation of ~0.025 cm$^3$ starting at ~2000 sec before slip onset (Fig. 5B). Assuming that the injected volume before slip primarily remains in the fault core, we can approximate the equivalent amount of fault dilation. For an elliptical fault with 100 cm$^2$ area every 1 ml (=1 cm$^3$) fluid volume change corresponds to ~0.25 mm of fault dilation, suggesting dilation of only 5 μm prior to dynamic rupture. Such small signals are detectable for high damage faults at $P_p = 5$ MPa, but low damage faults at $P_p = 0.5$ MPa exhibit no detectable dilational signal (Fig. 5B). The amount and duration of dilation is substantially reduced for all tests at $P_p = 15$ MPa

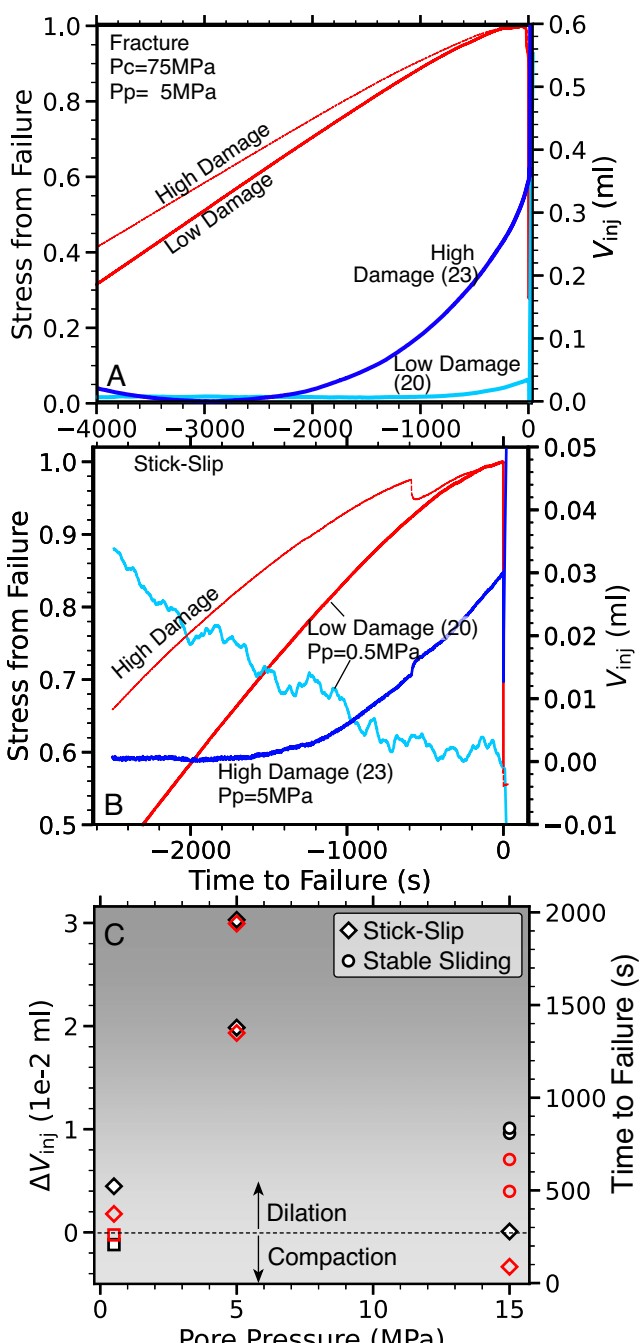

**Fig. 5 | Higher micro-crack damage amplifies sample dilation. A** Axial stress increase before intact-rock fracture at $P_p$ = 5 MPa (red curves, normalized by failure stress) and change in pore volume ($V_{inj}$) for sample 20 and 23 (see label in parenthesis) with low and high damage (cyan and blue curves). The sample with high initial damage undergoes pronounced dilation with pore volume change of -0.35 cm³. (Source data file provided) **B** Same as **A** but for stick-slip phase of experiment 20 and 23 (see label) at $P_p$ = 0.5 and $P_p$ = 5 MPa, again with much more significant dilation for the high-damage sample. Pore volume reduction for low damage sample is due to axial compression (Source data file provided). **C** Overview of pore volume change, $V_{inj}$ (black markers), and onset of dilation relative to time-to-failure (red markers) during stick-slip on faulted samples. The low damage test without dilation is highlighted by a square marker at the lower left.

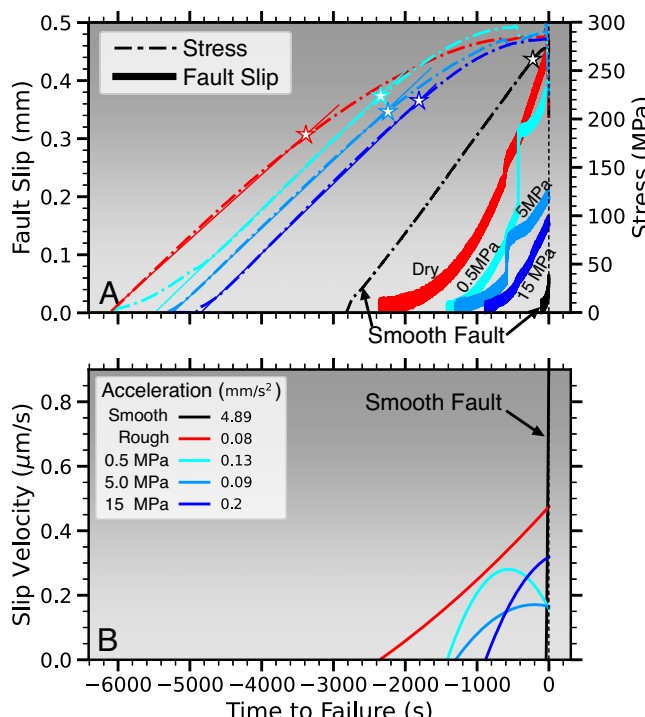

**Fig. 6 | Increased pore fluid pressures and lower fault roughness shorten the duration of premonitory slip and lead to rapid slip-acceleration toward failure. A** Differential stress (dash-dotted curves) and fault slip (solid curves) before stick-slip failure on dry (red) and fluid saturated faults (blue colors, see labels for pore pressure). Black curve shows brief premonitory slip on a smooth fault. Note that experiments with longer periods of premonitory slip also exhibit extended periods of non-linear stress increase before slip (start of non-linear stress is highlighted by colored stars). **B** Slip velocity evolution for the faults in **A**. (Source data file provided).

measured change in pore volume: 1) The pore volume increase prior to slip mirrors observations from rock fracture stages for which dilation and pore volume increase are a result of progressive crack formation, extension and coalescence. 2) The observed volume change is sensitive to the initial crack density and most pronounced for high-damage samples. We conclude that pore volume changes likely originate from changes within both fault core and damage zone.

### Preparatory processes and premonitory signals before failure

In the following, we analyze potential differences in preparatory signals (i.e., accelerating fault slip, seismic velocity reduction, AE event localization, AE rate increase and focal mechanism variability) with fault roughness and pore fluid pressure. Such signals are expected to stem from progressive strain localization within the fault core but also from distributed deformation across the fault damage zone. We focus on comparing measurements leading up to the first slip event on each fault to avoid effects from fault evolution and surface smoothing.

Fault slip and slip velocity before failure indicate a progressive shortening of the preparatory phase with increasing pore pressure (Fig. 6). Premonitory slow slip was estimated from strain gauge and vertical displacement measurements (see Method). Both stable sliding and stick-slip exhibit extended periods of slow, premonitory slip which approximately starts with pore volume dilation or earlier. The respective relative time periods expressed as percentage of recurrence intervals are 39, 23, 24, and 17% for rough faults at 0 (dry), 0.5, 5 and 15 MPa pore pressure, and 4 % for the dry, smooth faults (Fig. 6A). The shorter duration at higher fluid pressure is not solely an effect of differences in effective stress since peak stresses show only minor variations between the experiments.

which is also the regime that favors stable sliding over stick-slip (Fig. 5C).

Two observations suggest that not only gouge layer dilation but also crack opening across the fault damage zone contributes to the

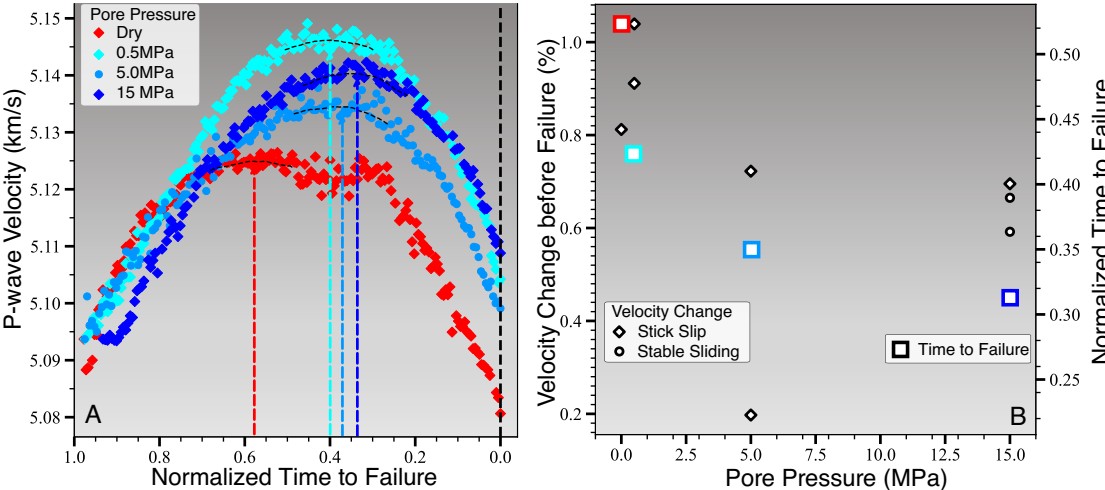

**Fig. 7 | Seismic velocity decrease starts closer to failure and is less pronounced in samples with higher pore pressures. A** Decrease in seismic velocity prior to failure in dry (red) and fluid saturated (blue markers, see legend) samples. Colored vertical dashed lines highlight the time of maximum velocity and black dashed line is the time of failure. **B** Relative velocity change (black circles and diamonds, see left legend) and average onsets of velocity reduction relative to time of failure (colored rectangles) as a function of pore pressure. (Source data file provided).

We observe that slow slip starts earliest on rough dry faults and much later, i.e., closer to failure on faults with higher pore pressures. Smooth surfaces exhibit the shortest duration of preparatory slow slip and most rapid acceleration whereas dry, rough faults exhibit the longest and most gradual acceleration toward dynamic failure (Fig. 6B). Increasing pore pressures partially bridge this gap, with more rapid slip acceleration for faults at $P_p = 15$ MPa. Similarly, differential stresses exhibit onsets of non-linear behavior that is progressively closer to failure with increasing pore pressures. Pre-failure stresses on dry, smooth faults are almost entirely linear until dynamic failure. Non-linear stresses before failure are expected to be a result of gradual fault activation, slip and microcrack damage evolution.

Seismic velocity changes before rock fracture and fault slip commonly exhibit two phases (Fig. 7): Initially velocities increase due to crack closures with increasing stress and then decrease due to microcrack formation closer to failure[12,19,24,31]. In our experiments, the reduction in seismic velocities starts between 1500 and 3200 s before failure, which is substantially earlier than the measured onset of sample dilation and slow fault slip (Fig. 7B). The drop in seismic velocity is most pronounced and starts earliest for the dry, rough-fault experiment. Fluid-saturated tests exhibit shorter periods of velocity reduction before failure with little difference between tests at 5 and 15 MPa.

Changes in AE sensor positions due to slow fault slip affect the velocity estimates to some degree but cannot explain the entire velocity drops before failure. These changes contribute up to 0.29% to velocity reduction at dry and low pore pressure conditions and up to 0.14% at high pore pressure conditions, which accounts for about 20–30% of the observed velocity drops before slip. These estimates are based on assuming that slip occurs localized on a single surface with a slip vector parallel to the $\sigma_1 - \sigma_3$ plane. Systematic differences in the duration of velocity decrease before failure are not affected.

The observed changes in seismic velocity are a result of crack closure and new crack formation with increasing deviatoric stresses. In addition to crack evolution, seismic velocities are sensitive to the degree of fluid-saturation across the fault damage and gouge zone[52]. Shorter periods of precursory P-wave velocity reduction are in approximate agreement with more rapid fault activation at elevated pore pressure.

We further investigate damage evolution before failure by examining AE event records across the 14-channel piezo-ceramic array. AE events occur consistently more localized around faults with high pore pressure (Fig. 8). This localization is observed after rotating all AE events into a best-fit fault coordinate system and tracking temporal changes in the 10th and 90th percentiles of across fault AE distances.

Preparatory processes before dynamic failure on dry faults produce repeating patterns of spatially distributed AEs at low stresses which progressively localize onto the fault (Fig. 8A). This pattern disappears at pore pressures of 5 MPa or higher and AEs stay largely localized throughout, potentially explaining the more modest relative P-wave velocity changes at these pressures (Fig. 8B).

AE rates are higher on dry compared to fluid-saturated faults especially early on during the loading cycle (Fig. 9). Seismic activity on dry and fluid-saturated rough faults accelerates more gradually which can be described by an exponential increase to failure. Dry, smooth faults, on the other hand, show very short duration precursory seismic activity and fast, power-law-like acceleration toward failure. Increasing pore pressure partially bridged the gap between rough and smooth dry faults; however, AE rates are highly variable within and between experiments (Supplementary Fig. S11). This variability may be characteristic for slip at transitional stresses between stable sliding and stick-slip where small perturbations can lead to substantial bursts in AE rates (Fig. 9A)[15,40].

Higher pore fluid pressures not only compress preparatory phases in time but also lead to more similar moment tensors. Moment tensor variability is determined from the distribution of minimum rotation angles of pairs of moment tensors within specific magnitude bins which is a proxy for geometric complexity at the scale of AE events (Fig. 9B)[39,53]. Overall, focal mechanism variability is systematically reduced at higher pore pressures (Supplementary Fig. S12) and respective moment tensors become more similar for large-magnitude events. Focal mechanism variability is further reduced on smooth faults which also show a similar relative decrease from small to large-magnitude events. We conclude that both high fluid pressure and low roughness lead to less geometric complexity, especially at the scale of the largest magnitude AEs.

## Discussion

Fault creep and abrupt failure in our lab experiments is preceded by an activation process that varies in duration depending on fault roughness and fluid pressure. This preparatory process is dominated by aseismic slip but also produces distinct patterns of high-frequency AE signals that systematically evolve toward failure[15]. Dynamic rupture on dry rough faults is preceded by extended slow slip and productive seismicity sequences which are controlled by large fault heterogeneity.

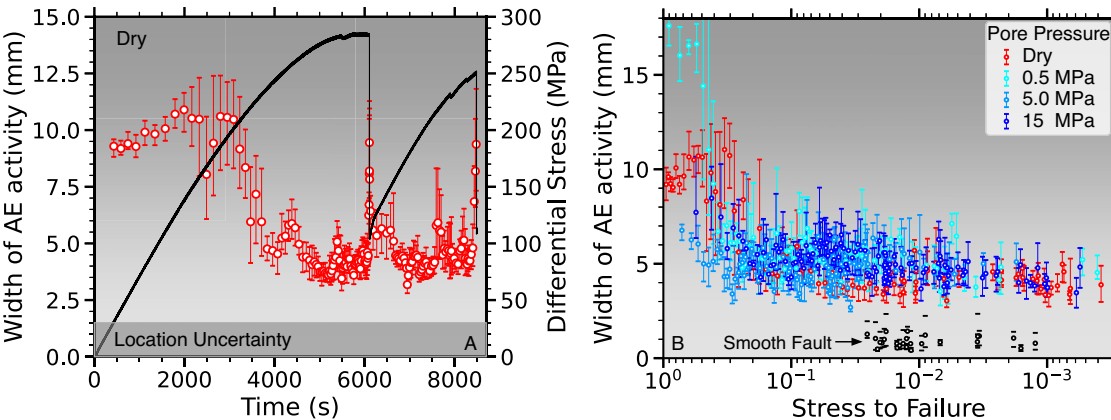

**Fig. 8 | Rough faults under dry and low $P_p$-conditions exhibit progressive localization of seismic events towards failure, which is largely absent at elevated pore fluid pressures. A** Width of AE activity zone defined by 10th and 90th percentiles of the across fault AE event distributions on dry faults (red markers, error bars show min. and max.). Differential stress is shown in black. **B** Width of AE zone for four experiments at different pore pressures (see legend) as a function of stress to failure error bars show min. and max.). The AE activity zone exceeds the location uncertainty (gray shaded region) in all cases. (Source data file provided).

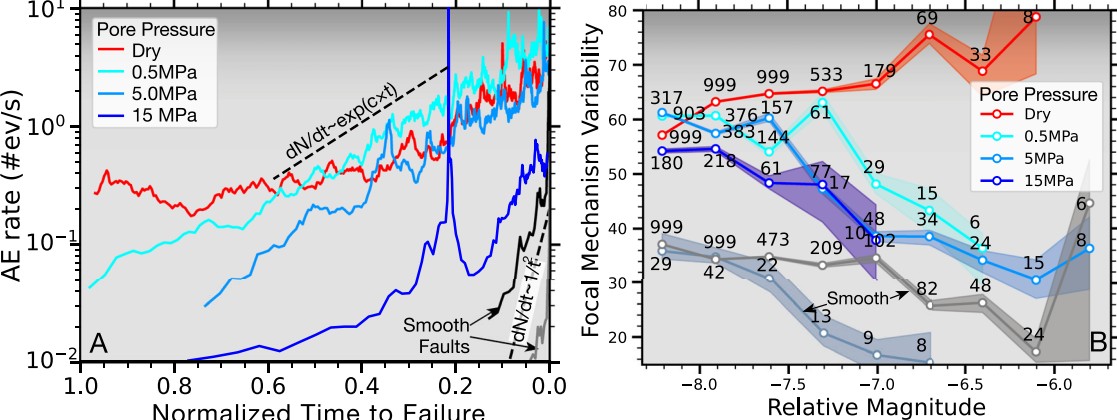

**Fig. 9 | Increasing fluid pressures lead to temporally compressed preparatory phases and reduced focal mechanism variability before slip. A** AE rate evolution leading up to failure for four tests at different pore pressures (see legend). AE rates on fluid saturated faults roughly exhibit an exponential increase (dashed line, see label) toward failure whereas dry, smooth faults (black and gray curves) show a much more rapid, power-law-acceleration (dashed line, see label) toward failure. **B** Median focal mechanism variability as a function of magnitude for dry rough (red) and smooth (gray) faults as well as for different fluid pressures (see legend). Shaded area are 95% confidence intervals and the numbers indicate how many focal mechanisms were used within each magnitude bin (see Supplementary Fig. S12 for complete distributions). (Source data file provided).

The early preparatory phase on this fault is dominated by compaction type events (i.e., negative isotropic moment tensor components) with increasing amounts of shear type events when approaching failure. Preparatory slip on smooth faults is much shorter and accelerates rapidly towards failure. Both surface smoothness and high pore pressure result in increased shear type events and more localized damage[54]. Notably, progressive damage localization is a hallmark of many previous experiments on dry samples[13,15,19,22], but is largely absent at high pore pressures.

Preparatory signals are more pronounced in samples with high geometric heterogeneity, e.g., due to high micro-crack damage and roughness. However, this effect also depends on confining pressure. During confining pressure increase from 75 to 150 MPa, we observe progressively more linear stress-strain relations even for high-damage samples but with respectively lower elastic moduli. AE event source types during stick-slip in high-damage samples are comprised of up to 9% more shear type events, are slightly less localized and exhibit higher peak AE rates compared to low damage samples. These observations suggest that the degree of structural complexity and damage may play an important role in amplifying precursory signals at low to moderate confinement and, potentially, also at low effective stress.

The apparent, macroscopically continuous slip-acceleration toward failure is accommodated by intermittent slip across populations of microcracks and frictional contacts (Fig. 4). These micro-scale processes are governed by pore-pressure, grain size (or roughness) and temperature which control the rheology of the granular gouge zone[55]. The resulting effective slip velocity across the gouge layer is a function of real area of contact ($A = (\mu_0\bar{\sigma}/\chi)(d/d_0)^{(m/n)}$) which controls yield strength, $\sigma_y = \chi A$ of the asperity populations, where $\chi$ is contact hardness, $\bar{\sigma}$ is effective normal stress, $\mu_0$ is friction, $d/d_0$ is average contact size relative to a reference contact size and the exponents $m$ and $n$ govern the degree of non-linearity between normal stress and contact size changes[3,55].

A respective constitutive law for asperity distributions can be formulated similar to crack distributions in Eq. (1) but with the added terms for grain size and temperature dependence that capture rheological effects within the gouge layer[35]:

$$V = V_0 \left(\frac{\tau}{\mu_0\bar{\sigma}}\right)^n \left(\frac{d}{d_0}\right)^{-m} \exp\left[-\frac{Q}{R}\left(\frac{1}{T} - \frac{1}{T_0}\right)\right] \qquad (2)$$

where $\tau$ is shear stress, $\mu_0$ is the reference friction at velocity $V_0$; $T_0$ and $T$ are reference temperature and temperature during frictional sliding and $Q$ and $R$ are activation energy and universal gas constant. Effective slip velocity increases as a power-law with increasing stress to failure. This non-linear behavior is expected to also depend on heterogeneity with more homogeneous samples leading to larger power-law exponents[33].

The constitutive law suggests that higher pore pressures lead to a more rapid non-linear increase in slip velocity by reducing effective normal stress, $\bar\sigma$, which is also observed here. In addition, the presence of fluids modulates several fault thermo-dynamical and rheological processes such as thermal pressurization, asperity creep, pressure diffusion as well as stress corrosion and sub-critical crack growth[3,43,56]. Temperature is a predicted catalyst for these processes thereby also affecting foreshock generation, which should be explored in future lab tests.

The above constitutive relation predicts that fault slip velocity is strongly affected by the real area of contact, $A$, with larger values of $A$ hindering more rapid slip propagation. Consequently, slip acceleration before dynamic failure may be associated with a progressive reduction in real area of contact. This area also evolves with subsequent stick-slip due to wear production, grain comminution and alignment of phyllosilicates. Similarly, shorter preparatory phases and rapid slip acceleration on smooth faults are explained by smaller contacts and more rapidly changing real area of contact compared to rough faults with larger $d$.

Pore fluid pressures reduce the effective stress and cause a moderate increase in damage zone stiffness, which promotes stable sliding and increased rupture nucleation patch size[12,57]. A simple empirical criterion for slip instability can be formulated by balancing elastic unloading rate and frictional strength reduction with fault displacement[58]. Stick-slip is thus expected if the following inequality is satisfied[2,57]:

$$K < \frac{(b-a)\bar\sigma}{D_c},\qquad(3)$$

where $K$ is system stiffness, $b-a$ are frictional rate-state parameters, $\bar\sigma$ and $D_c$ is the roughness-dependent characteristic weakening distance. A condition for stick-slip within this framework is rate-weakening friction (i.e., $b>a$). Stiffer lab samples and loading machines increase rupture patch nucleation size and promote stable sliding. In addition, stable sliding may be caused by pore pressure effects on normal stress and frictional parameters[46,47].

We observe a large difference between heterogeneous rough surfaces with long preparatory phase and homogeneous, smooth surfaces with rapidly accelerating slip. The effect of surface roughness on the development of foreshock sequences may depend on the relative spatial scales of respective seismic events and average separation-distance between load-bearing asperities. The smallest AE events have rupture dimensions of ~0.5 mm (and average size of ~1 mm) based on seismic moment and corner frequency, which exceeds the respective corner wavelength of surface topography on smooth faults (i.e., ~0.1 mm)[39,59]. Consequently, the saw-cut surface can be considered smooth (or planar) at the scale of most AEs and small-scale heterogeneities provide only limited barriers for AEs which grow and coalesce rapidly towards dynamic failure[17]. Rough surfaces, on the other hand, appear self-affine up to a wavelength of ~3 mm[39] which is similar to the scale of the largest AEs[59]. Such large-scale heterogeneity and separation length between asperities provide efficient barriers that can stop AE ruptures. The corresponding AEs likely comprise the more extensive foreshock sequences on rough faults[60].

The difference in preparatory processes on rough vs. smooth faults is partially bridged by tests with increasing pore pressure on rough faults. A potential cause for these pore pressure effects is the relative reduction in geometric and stress field heterogeneity. We hypothesize that fault contact stresses progressively evolve to a more homogeneous state due to either higher normal loads, fault slip, or pore pressures.

Foreshocks and precursory signals are observed before most failure events in lab experiments[12,13,19]. Consequently a key question is why such precursory signals are largely absent in nature. Several factors may contribute: 1) Instrumental resolution-limits prevent the detection of nucleation processes which may occur at millimeter scales[50,59,61]. 2) Precursory activity may vary substantially between different regions and observations in one place may be different in other places[4,5,62]. 3) Variations in stress over series of stick-slips in the lab may significantly exceed stress changes in nature[14].

Our results provide an additional explanation, i.e., differences in fault heterogeneity which strongly affect preparatory processes and associated seismic signals. The here explored stress conditions correspond to shallow to mid-crustal depths. Larger depths likely lead to further compression of the nucleation period and rapid acceleration toward failure. The shortening of preparatory phases is a result of both stress and pore pressure increase with depth. If the here reported trends continue at pore pressures representative of seismogenic depth, we expect significant temporal compression of foreshocks and rapid slip acceleration toward dynamic failure in nature. Lower roughness and increased normal stress are expected to reduce the nucleation length of unstable ruptures leading to rapid slip acceleration and short foreshock periods[23].

The effect of different crustal conditions on foreshocks is qualitatively illustrated in Fig. 10 where high-damage as well as low pore and confining pressures at shallow depth are associated with the longest foreshock activity.

The trade-offs between these parameters in nature are difficult to assess, however our lab results suggest that both fault roughness and pore pressure have substantial effects on microseismicity characteristics. Future experiments would benefit from testing a wider spectrum of fluid pressures, temperatures and stresses to confidently identify the most important parameters.

This study suggest that earthquakes on immature, rough faults under drained conditions, e.g. close to the earth's surface, are more likely to generate productive foreshock activity. The spatial dimension of such foreshocks may be determined by the scale of heterogeneity, e.g., roughness. For instance, the generation of foreshocks of $M_w = 1$ with a stress drop of 1–10 MPa may require geometric or stress field heterogeneity at the scale of 10s of meters. An example of a shallow, productive foreshock sequence that lasted for about two years is shown in Fig. 10B. The events leading up to the 2016, M5.1 Fairview earthquake in north-western Oklahoma were driven by induced poroelastic stresses due to field-wide injection at large distances[63]. Shallow tectonic earthquakes may be associated with similar foreshock sequences under comparable conditions.

In conclusion, we demonstrated that elevated pore fluid pressures and reduced roughness lead to more rapid slip acceleration and shorter precursory periods. Such shorter periods are caused by less heterogeneous fault stresses and lower geometric complexity. Homogeneous faults remain locked for longer during the interseismic period and subsequent activation processes, i.e., stable sliding vs. stick-slip, are affected by inherited structure and the degree of strain localization. Focal mechanisms of AE events at high pore pressure are increasingly dominated by double couple components - analogous to natural seismicity. Rapidly accelerating nucleation processes at large pore and confining pressures may be difficult to detect outside of high-resolution lab studies. Foreshock activity may be amplified by geometric complexity (e.g., damage and roughness) and stress heterogeneity, which is more likely observable on immature, shallow faults in nature.

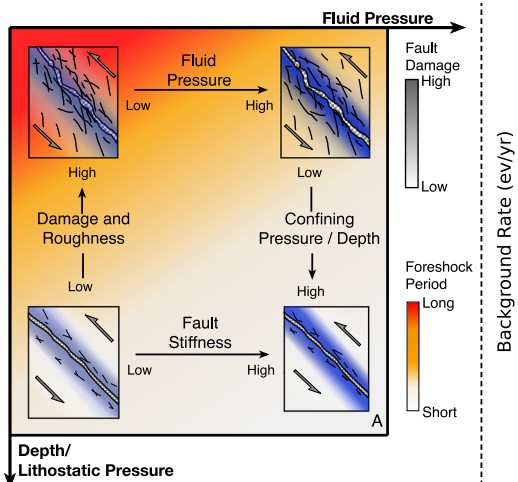

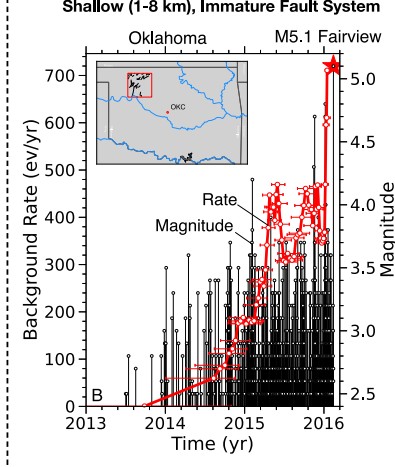

**Fig. 10 | Shallow depth, high-damage, low damage-zone stiffness and low fluid pressure lead to long preparatory periods with pronounced foreshocks before failure. A** Schematic illustration of how pore pressure, damage, damage zone stiffness, and confining pressure affect foreshock duration in laboratory tests. Consequently, foreshock activity is more likely observable in nature before shallow earthquake, e.g., in Oklahoma. **B** Example of a shallow M5.1 earthquake (red star) near Fairview, Oklahoma, preceded by seismicity with increasing rates (red curve, error bars show time window used to calculated background rates) and magnitudes (black markers). Map was created with the matplotlib-basemap library for Python).

## Methods

### Experimental design and sample preparation

Triaxial compression experiments were conducted on 10 cylindrical (diameter = 50 mm, height = 105 mm) Westerly granite samples at confining pressures, $P_c$ of 75 and 150 MPa during sample fracture and subsequent frictional sliding (Supplementary Fig. S1, Supplementary Table S1). The experiments include three tests at dry conditions (two on smooth and one on a rough fault) and seven tests on fluid saturated samples with incipient faults that evolved from fresh fractures. We generated two-types of fault roughness by introducing 30° saw-cuts and fracturing initially intact rocks at $P_c = 75$ MPa. The natural fractures were guided by two 25 mm notches on opposite sides of the samples, inclined at 30° to the loading. The samples were subsequently reloaded at $P_c = 150$ MPa leading to stick-slip or stable sliding. Fault roughness variations were examined in X-ray computer tomography and white-light interferrometry scans, yielding about two orders of magnitude difference in rms-roughness between smooth and rough faults[39].

All fluid-saturated tests were conducted at constant pressure boundary conditions facilitated through a central borehole (Supplementary Fig. S1), to simulate the effect of a fault patch in nature, embedded within a hydraulically connected fault damage zone. Such conditions are reasonable if near-fault damage is high, leading to increased fault permeability parallel to the slip plane[64]. Pore pressure conditions are expected to transition from quasi-drained (i.e. constant pressure) to undrained (i.e. constant volume) for natural and lab faults at time scales of rapid crack and rupture propagation. In our experiments, such a transition leads to substantial pore pressure fluctuation within the borehole due to the interplay of pore volume changes and hydraulic servo control, leading to initial pressure drop during pore volume increase and subsequent pore pressure spikes after failure (e.g. Supplementary Fig. S9).

Prior to sample fracture, we saturated the faults for more than one hour by injecting fluid through a central borehole. We observe systematic P-wave velocity increase which approaches a constant value when pore pressures homogenize (Supplementary Fig. S1). The central borehole serves as direct hydraulic connection between fault and servo-controlled hydraulic pumps, allowing us to maintain constant pore pressures on the fault prior to dynamic failure. We use the hydraulic system to keep track of fluid volume changes within the sample ($V_{inj}$), which is a proxy for volumetric changes of connected pore space.

We simulate the effect of elevated fault damage by heating samples 21–26 to 700 °C leading to pervasive microcrack damage caused by different thermal expansion of the various granitic mineral phases. The resulting microcrack damage results in strongly non-linear stress-strain relationships at low confining pressures and notable difference in bulk moduli, with thermally treated samples exhibiting a reduction in bulk modulus by a factor of ~2–3 (Supplementary Fig. S2).

All samples were separated from the confining oil by an elastic rubber jacket and loaded axially at a displacement rate of $3 \cdot 10^{-4}$ mm/s up to a maximum vertical displacement of $u$ ~6 mm. An external load cell and two vertical and horizontal strain gauges measured axial force and strain at sampling rates of 10 Hz.

### Empirical stiffness estimates

The utilized loading rig at GFZ-Potsdam includes several steal and Teflon disks to facilitate fluid injection and to minimize end-cap friction during stick-slip, all of which affect system stiffness, in addition, to variable rock sample stiffness with varying damage. As a consequence, we estimate the axial system stiffness, $K_{ax}$, which is the sum of the stiffness of all load-bearing components, $N$ (with $K_{ax} = \sum_{i=1}^{N}(1/K_i)^{-1}$) empirically during elastic loading and unloading[65]. For this purpose, we fit the linear components of the early stress-displacement curves and average loading which exhibit comparable values at $P_c = 150$ MPa (Supplementary Fig. S4). As expected, the system stiffness is lower in faulted compared to intact samples and is always much lower than the machine stiffness of 350 MPa/mm (Supplementary Table S2). Notably, empirical system stiffness show little sensitivity to the presence thermally-induced micro-cracks at least during axial loading at high confining pressure (Supplementary Fig. S4).

We observe a moderate stiffness increase with larger pore pressures. This increase in stiffness is approximately equal to pore pressure increase (i.e., for $\Delta Pp = 15$ MPa, $\Delta$Stiffness = 15 MPa/mm), however there are also large deviations from this trend. Additional super-imposed effects stem from fault stiffness variations due to inherited fault geometry from the natural fracture process, fault evolution and damage production with increased slip.

## Seismic velocity, acoustic emission detection and locations

We use pairs of uni-polarized piezo-ceramic transducers as active ultrasonic pulse emitters and receivers and determine P-wave velocity from known transducer distances and measured travel times. For this purpose, we digitally record the onset of pulse generation at the sending transducers and automatically pick phase arrival times at the receiving transducers, using an autoregressive AIC picking algorithm applied to waveforms trimmed close to the expected P-phase pick[66].

We analyze passive seismic sources across a 16 channel, high-speed data acquisition system[67]. Accurate AE locations were possible due to high-quality automated picks, high sampling rates (10 MHz) and time-dependent, anisotropic, layered velocity models from ultra sonic pulses emitted every 30 s. Seismic velocities can vary by as much as 10% during the fracture of intact rocks and only 0.2 to 1.2% prior to stick-slip in faulted samples. We searched for AE events within 100 μs time windows, and kept events with high SNR, at least 8 station picks and travel time residuals of less then 0.5 μs, thereby minimizing the likelihood of erroneous detection and locations. AE location uncertainty was between 0.5 and 3 mm[21].

We determined relative AE event sizes by averaging peak amplitudes across the laboratory array of piezo-ceramic transducers and correcting for source-station distances. We compute local magnitudes based on peak AE amplitudes averaged across the entire array and corrected for source receive distances[68]. We assign a minimum event size corresponding to the smallest grain size at the sub millimeter-scale[59].

## Acoustic emission source types

We determine AE source types from averaged first motion polarities across the 16 channel array with 3D configuration around the sample which provides reliable coverage of the focal sphere. Average source polarity, $P_{ave}$, are computed for each event, using[68]:

$$P_{ave} = \frac{1}{n} \sum_{i=1}^{n} \text{sign}(A_i) \qquad (4)$$

where $n$ is the total number of AE transducers that were used to locate the event, and sign($A_i$) is the polarity recorded at each channel. Based on $P_{ave}$, we separate events into dominant compression (C-type with $P_{ave} < -0.25$), shear (S-type with $-0.25 \leq P_{ave} \leq 0.25$) and tensile source components (T-type with $P_{ave} > 0.25$).

We compare the simple average polarity measures with full moment tensors determined from first motion amplitudes, using the *hybridMT* package[69]. Before the inversion, $P$-wave amplitudes were corrected for coupling effects between AE sensors and rock surfaces, which change as a function of confining pressure[54,69]. The corrected amplitude data were inverted for six independent moment tensor components using a least squares approach[69]. The results provide a more detailed insight into source processes of individual events (Supplementary Fig. S5) while average behavior of event population is consistent between polarity estimates and moment tensor decomposition (Figs. 2 and 3)

## Fault slip velocity estimates

Fault displacements and velocities were determined by comparing far-field axial displacement measurements with vertical strain gauge measurements on the sample. We determine the total strain across the fault by rotating the residual strain from total compression minus machine and sample deformation into a fault parallel coordinate system. Residual vertical strain, $\gamma_{res}$, attributed to fault slip is determined from:

$$\gamma_{res} = \gamma_{ax} - \gamma_m - \gamma_s, \qquad (5)$$

where $\gamma_{ax}$ is total axial compressive strain determined from an LVDT attached to the loading pistons, $\gamma_m$ is deformation of the loading machine, and $\gamma_s$ is sample deformation. It should be noted that the resulting displacement, $D_f$ and velocity, $v_f = dD_f/dt$ estimates are upper bounds because residual strain may not solely be accommodated by localized fault slip but also by distributed deformation across the fault zone.

## Microstructure imaging

Post mortem microstructure which developed due to the cumulative effects of dynamic fracture and frictional sliding were analyzed in X-ray computer tomography scans and optical and scanning electron microscopy images. Each sample was carefully removed from the pressure vessel and kept in the neoprene jacket in order to preserve the generated deformation structures. A micro X-ray computer tomography scan (GE Phoenix X-ray nanotom 180 NF, operated at 140 kV and 100 μA with a voxel size resolution of 30 μm) was then performed. The resulting images resolved the overall fault geometry specifically shear fracture formation between the two notches and connection between fault and bore. The fault zone exhibited several undulating strands inclined at an average angle of 30° to the $\sigma_1$ stress axis.

After vacuum impregnation with blue colored epoxy resin, thin sections of the samples were prepared perpendicular to the generated fault zone for further microstructural imaging by optical and scanning electron microscopy (FEI Quanta 3D Dual Beam, operated at 20 kV at 10 mm working distance). In general, the observed microstructures indicate that the connection of the notches by shear fracture coalescence, as well as single stick-slip events and stable sliding, were predominantly accommodated by brittle processes including microcracking, cataclasis and granular flow. The internal structure of the generated fault zones is complex and consists of various structural elements that were observed in all samples comprising a damage zone, which surrounds a fault core that contains multiple localized slip surfaces filled with gouge.

## Nearest-neighbor clustering and background seismicity rates

We separate the record of earthquake locations, origin times and magnitudes in California (Fig. 1) and Oklahoma (Fig. 10) into clustered and background events. This separation is based on nearest-neighbor event-pairs which are determined from spatial-temporal distances of event pairs scaled by parent event magnitude[70]. The nearest-neighbor distances are used to detect background events at large distances and times and determine respective background rates. The observed nearest-neighbor distance distributions are compared with randomized Poissonian catalogs that have the same number of events and magnitude distributions as the original catalogs. Events are categorized as clustered at distances and times below the 99th percentile of the randomized catalogs.

## Reporting summary

Further information on research design is available in the Nature Portfolio Reporting Summary linked to this article.

## Data availability

The generated laboratory data and figure source data are provided as Source Data file at: https://doi.org/10.6084/m9.figshare.24894705. Southern California seismic data in Fig. 1B can be accessed through the Southern California Earthquake Center: SCEDC (2013): Dataset. https://doi.org/10.7909/C3WD3xH1. Oklahoma seismic data in Fig. 10B is available at: https://ogsweb.ou.edu/eq_catalog/[71].

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

## Acknowledgements
The authors would like to thank Stefan Gehrmann and Matthias Kreplin for rock sample preparation at GFZ-Potsdam, Germany. This research was made possible by a Humboldt fellowship and NSF Career Award to T.G. (Award Number: 2142489).

## Author contributions
T.G. conceived the study and performed the laboratory experiments. Data analyses was performed jointly by T.G., G.K., K.P. and V.S. The manuscript was written jointly by T.G., G.K., V.S. and G.D.

## Competing interests
The authors declare no competing interests.
