## [Peer Review File · Nature Communications]

A laboratory perspective on accelerating preparatory processes before earthquakes and implications for foreshock detectabilityREVIEWER COMMENTS

Reviewer #1 (Remarks to the Author):

I have reviewed the paper entitled "Why are foreshock rare in nature? A laboratory perspective" by Goebel et al., as a potential manuscript to be published in Nature Communication. The authors report on laboratory experiments where they investigate the effect of increasing fluid pressure on the stick slip behavior of granite samples. They used three different sample configurations: 1) dry saw-cut smooth surfaces; 2) in situ fractured samples and 3) the same as in 2 but thermally cracked to understand the interaction of changing stiffness and hydraulic properties on the mechanical behavior. To inform the mechanical data they use an array of PZTs to retrieve the ultrasonic wave velocity (active source) and passive Acoustic Emission (AE). From the experiments and AE data emerge a very complex interaction between the main fault zone and the surrounding rocks as a function of fluid pressure that is also highlighted by the AE.

I find the idea behind the paper very interesting, however, due to the large amount of data I got lost in understanding what is the real focus of the authors. As the title reads I would have expected a different type of paper. Here the authors mainly focus on the effect of fluid pressure on fault slip behavior so that I think the title is way too general since not all faults are overpressurized.

In general the paper is well written, even though I found many typos throughout the manuscript. I think that the authors should work on reorganize and clarify their data selecting what is really important for the message they want to send. Most of the time more data can only confuse the reader if there is not a clear explanation of the meaning of those data. In fact at the end of the manuscript I found myself confused on what data really supported their conclusions. Nonetheless, the data are very interesting and the methodology sounds good.

In conclusion I think that the paper deserves publication however after some very major revisions and reorganization of the manuscript. I will list a series of comments below that I hope will help in improving the manuscript.

Sincerely
Marco Scuderi

Line-by-line comments:

L34: why use Leeman as a reference in this context?

L34-36: I do not find myself in agreement with this statement since the failure of intact rocks involve mechanical processes that are quite different from frictional sliding along pre-existing faults with gouge. Indeed, here it arises a problem that I found throughout the paper and I invite the authors to clarify. In their opinion is faulting (i.e. reactivation of a preexisting plane of weakness) a fracture of frictional processes? I read in the text jumping from one to another and this does not help the clarity of the paper.

L45: crack alignment, can the authors be clearer on what they mean with this process? Is this coalescence on microcracks in an intact rock?

L59-61: stress corrosion processes are quite different from subcritical crack growth. Can the author be more specific?

Line 82: on the fault structure there are more references to add.

L84: here the authors talk about frictional break-down processes that are different from fracture propagation as invoked before. Can the authors clarify which constitutive framework they want to use? There is nothing wrong to treat both (fracture and friction) but the authors should be clear about it.

L106: here the author mention a confining pressure of 120MPa but I do not see experiments at this pressure from table S1

109-111: can the author mention that the natural fracture was induced by notch on the sample. I found this information buried in the methods but I think it is important to mention here to help the reader understand the comparison with saw-cut.

L121-122: "... lower bulk moduli and higher across-fault crack densities for thermally treated samples". I do not understand this sentence. Is higher for increasing?

127-128: about the surface smoothing. The authors here refer to figure 2 where naturally fractured rock are reported. However, also based on the microstructures, I have some doubts in the interpretation of surface smoothing since the authors show the creation of gouge that should at this point control fault mechanics and the characteristics of stick-slip. Why the authors highlight surface smoothing and not gouge creation associated with strain localization?

Figure 2: Coupling this figure with table S1 I see that there are no experiments at high fluid pressure and "normal" granite samples. I see only the experiment 24 that arrives at the fluid pressure reported in fig 2c. However this is thermally damaged but I do not see specified anywhere in the caption. Can the authors clarify? Also, why do they compare experiments with different boundary conditions? As mentioned in the text the thermal cracking changes the elastic moduli of the rock so that stick slip behavior should be different from the "normal" intact rock.

L130-131: this statement should be clarified and contextualized. Clearly the presence of fluids influence the microscale deformation via chemical reaction at the grain scale, however, I am confused on the reasoning of why high fluid pressure should increase small scale deformation. Indeed, since fluid pressure is isotropic in all direction and lower the effective stress in my opinion it should decrease stress at the contact scale possibly reducing the real contact area and doing so inhibit shear deformation.

L134-138: I find these sentences a little confusing since it is not clear to me if the authors are referring to intact rock deformation or gouge. The authors should be more clear about it and maybe use a different terminology. Pore space collapse is usually referred to intact rock deformation after Yielding at P^* (i.e. Zhang et al., 1990). In the context of gouge deformation cataclasis and grain comminution are processes that control porosity. However, when talking about gouge deformation differences should be made between high strain and low strain domain since porosity will be heterogeneous.

Figure 3: the authors use this figure to show that shear type events are dominant. I would be curious to know where those events come from, if they are located in the damage zone or within the sample and so controlling shear deformation and stick slip. In the methods I read that the AE location uncertainty is between 0.5 to 3mm and from the microstructures in figure 4 I see that the gouge is about 100micron thick. At line 139 to 156 the authors seem to relate these AE to the microscale of gouge deformation. How can they be sure that the AE are representative of microscale deformation at the contact scale of the gouge? I would also be curious of the central frequency of the piezoelectric sensor they use. Is that high enough to resolve such detail?

L160-163: I think that this information should go much earlier in the paper since it is a great importance. In fact if 2 out of three experiment show stable behavior, why the authors show only the experiment that show instability? How much those instability are producible?

L166-167: to me it is very counterintuitive the fact that fluid pressure stiffen the rock. In fact, some measurements we made in our lab on sandstones (unpublished) shows the opposite. Looking at the paper the authors refer to, the stiffening seems related to crack density. If this is true does it mean that the sample that show stick slip has much less crack density? The authors report figure S4 as a measurement, but since they also have P-wave velocity, do they see a stiffening from such measurements?

L228-234: I would like to give another perspective on these data. Since fault porosity is quite low, looking at the microstructure, I think that the pre-seismic fluid volume increase can be due to the opening of interconnected fractures in the damage. Indeed, high damaged fault is the one that show this signal. If such dilation was happening in the gouge, high and low damage should hav

shown the same amount of dilation since the gouges look very similar in porosity.

L237-245: I agree with this interpretation.

L250-251: However, it seems that increasing fluid pressure also decreases the duration of the entire seismic cycle. I think that to make a direct comparison the authors should express the duration of pre-seismic creep as a percentage of recurrence interval. The different recurrence time it means that fluid pressure also affects frictional healing that can generate a different premonitory stage.

L307-308: To have the definitive proof if this is true the authors should compare rough dry faults at the same mean stress as the high fluid pressure. Could it be due to a reduction of effective stress?

L347-348: I think this is a little bit of an overstatement in particular for the micro-structural characteristics.

L376-382: Can the author clarify this statement. In my opinion, larger contact area should inhibit accelerated creep and unstable behavior. If we think to the combination of adhesion friction theory and the slip weakening law, in order to have a stress drop and slip acceleration the contact area must decrease as a function of slip. Indeed, many experimental works can be found on phyllosilicate rich faults where seismic slip is suppressed by the alignment of phyllosilicate foliae that increase the contact area, in some cases the contact area saturates and is expressed by a zero value of the rate parameter b in the context of rate and state friction.

L393-395: However, elevated fluid pressure can also change the critical rheologic stiffness $k_c = (b-a)s_n/D_c$ by decreasing the effective stress and push the fault to stability. In my opinion this is a more plausible mechanism.

L396-401: The information that aseismic creep has premonitory signals is the first time that it is mentioned. Also slow slip events are mentioned here for the first time. Can the author rework this entire section since I do not understand what is the focus and the message they want to send.

L424-425: Can the authors motivate this statement? Fluid pressure is an isotropic pressure that should act equally in all directions. So why it should act as a smoothing factor? What the author says it might be possible but under near lithostatic condition, while, from a rapid calculation here the authors are under sub hydrostatic condition (i.e. $\lambda \sim 0.2$)

L429-431: This sentence is quite confusing because there is at least a contradiction, how can normal load increase if also fluid pressure is increasing?

L445-446: I agree regarding the confining pressure, however, how can the author say it about the fluid pressure?

Methods:

L491: I think that more specification on how the fracturing was performed should be given here.

L493-495: where can I see these data?

L497-498: can the authors clarify this sentence? Is there more than one patch?

Reviewer #2 (Remarks to the Author):

Summary

This work shows why foreshocks are relatively rare in nature than the observations in the laboratory. They provide laboratory experiments that show the richness of laboratory foreshock is strongly dependent on fault pressure. They investigate the multiple factors that might influence foreshock – roughness, pre-existing damage, and pressure. They found even a small magnitude of

pressure can significantly reduce the preparatory phase of lab earthquakes. The paper further investigates the microstructural behavior of fault zones. The authors claim that the existence of pore pressure might be the reason for the relative absence of foreshock in natural fault systems.

Review Summary

I agree that foreshocks are more abundantly observed in the lab than actual earthquakes. As the authors discussed, I thought the paucity of foreshocks might have originated from the detection threshold. But now I am convinced that, as this paper says, the rock-fluid interaction might also influence the preparatory phase of natural earthquakes. The experiments seem to be fairly well-designed, and the analyses are convincing. I recommend publishing this paper.

Minor Comment

Table S1 shows the experimental setups for each 10 experiments conducted. However, it is hard to recognize which experiment is shown in the figures except figure 3. For example, figure 2 shows three experimental results but does not show which experiment they are. It would be helpful if authors could put an ID for each experiment shown in plots wherever possible, as done in figure 3.

Reviewer #3 (Remarks to the Author):

Summary:

Goebel et al. report on a series of laboratory experiments designed to understand nucleation processes leading up to laboratory stick-slip instabilities. The authors test the influence of fault roughness, damage, and pore-fluid pressure on nucleation processes, foreshock activity, and elastic

wave properties. They focus most of their analysis on the role of roughness and pore-fluid pressure.

One of the main takeaways from the work is that fluid pressure, roughness, and damage all act to modulate nucleation processes and potentially precursory signals. Specifically, enhanced stresses and pore-pressures seem to reduce precursory signals which may explain the lack of such signals in the Earth. Overall, I find this topic and set of experiments very interesting. I believe this work is suitable for publication once the authors address the following comments/questions below.

Major points/comments:

1. The authors show that increasing pore-fluid pressure decreases AE localization and reduces pre-seismic fault creep preceding failure. The authors also show that V_p decreases prior to failure with and without the presence of fluids. The authors claim that the duration and the overall magnitude over which V_p drops decreases with increasing pore-fluid pressure. I have a hard time observing these trends in the data in Figure 7. Based on the plots it appears that V_p decreases prior to failure, irrespective of pore-pressure. The magnitude and duration of the drop in V_p do change for pore-fluid pressures between 0-5 MPa, but for pressures > 5 MPa they are independent of pore-pressure magnitude (Figure 7B). I think this point is worth clarifying and acknowledging in the text since one of the major claims of this work is that increasing pore-fluid pressure reduces precursory signals.

2. I found it difficult to keep track of the varying boundary conditions (e.g., changes in damage, pore-pressure, and roughness). It could help the flow of things if these were separated more clearly in the text and figures. For example, right now Figure 2 discusses the role of pore-fluid pressure and then Figure 5 transitions to fault damage state and then there is a transition back to pore-pressure with Figure 6. And the effect of roughness is embedded in there as well.

3. The effect of damage on precursory acoustic signals is not discussed. Were AE and Vp data collected for these experiments? It seems like this would be a useful thing to discuss (if there are indeed acoustic data that go along with these experiments) because they play an important role in the main conclusion of the paper. Specifically, the effects of damage and the role it plays in generating (or reducing) precursory AE and Vp signals could be elaborated on in the discussion.

Comments:

L34: Not sure if Leeman et al., 2016 should be cited here. Normal stress dependence of b-value was shown in the work of Riviere et al., 2018 and also differential stress in Scholz, 2015. Yamashita

et al., 2021 also looked at surface roughness on 1-meter faults.

L36: Could also cite Scholz, 1968.

L45: Also strongly influenced by fault unlocking and slip rate.

L47: Can you clarify what you mean by short-term signals?

L56: I think this should be Johnson et al., 2017; Hulbert et al., 2019 also showed that various aspects of slow and fast stick-slip could be predicted using AE data.

L57: Can the authors comment on how they might think this model of crack propagation and coalescence would extend to cases where there foreshocks and AEs are generated from microscale processes occurring in a finite layer of fault gouge.

L78: See also Affinito et al., 2023; Samuelson et al., 2008; Proctor et al., 2020.

L115: What is the roughness of the smoothed surfaces compared to the surfaces that are fractured in-situ?

L124: Can you clarify what you mean by micro and macro scales?

L136: Do the data in Figure 2 correspond to the "rough" faults that were fractured in-situ or the smooth faults that were surface polished? Please clarify.

L139: Comparing Figure 2B and 2C one can tell that the number of shear type events has decreased

with increasing fluid pressure. However, this trend is more challenging to observe when comparing the pore-pressure increases within the same experiment in Fig 2C. The same can be seen in regards

to the tensile type events. Can the authors comment on why they think this trend is masked in Figure 2C? Is this somehow linked to differences in fault roughness? Is the effective normal stress held constant between the three experiments shown in figure 2 or does it change accordingly with pore-pressure?

L163: This is consistent with rate-state theory and previous lab experiments from Leeman et al., 2016; Scuderi et al., 2016;2017.

L166: The critical stiffness, K_c , will also change with effective normal stress as pointed out above. Also possible that D_c , and $a-b$ change with normal stress (Scuderi et al., 2016).

Figure 5B- Are the high and low damage experiments conducted at the same pore-pressure? For the low damage case, why is the fluid vol. decreasing? Is the sample compacting during this phase?

L251/Figure 6: Why do the slip displacement curves start near the end of the seismic cycle as opposed to the beginning of the seismic cycle? And are the stress drops the same for all events shown in Figure 6? Or does stress drop decrease with increasing pore-pressure? Is it possible that variations in roughness are playing a role in the pre-seismic changes in pore-pressure? Or do the events in Figure 6 correspond to the same shear strain/cumulative offset? It would be useful to plot

the pre-seismic change in fluid volume as a function of pre-seismic slip. Presumably you would find that these scale systematically with one another? Also, I suppose that part of the reason you have more pre-slip with decreasing pore-pressure is simply because the recurrence interval of the seismic cycles are longer from higher effective stresses and more frictional healing.

L261: Shouldn't the onset of inelastic creep be determined from a stress vs strain plot as opposed to stress vs time? Also, it seems that the onset on inelastic creep is not very sensitive to changes in

pore-pressure. The location of the stars are \sim in the same location on all three stress curves. It would be useful to extend the slip displacement curves back in time so that they encompass this deviation from linear-non-linear behavior.

L271: See also Shreeharan et al., 2020;2021. Scuderi et al, 2016

L287: what are the uncertainties in the AE locations? Please clarify.

L294-296: This is similar to what Bolton et al., 2023 observed for slow vs fast slip stick-slip events. The AE localization effect preceding failure is also seen the experiments from Marty et al., 2023.

L338: The AE localization does indeed seem to disappear with the addition of fluids. However it's interesting that V_p decreases prior to failure, irrespective of the pore-fluid pressure. Do the authors

believe this drop in V_p is related to shear/damage localization leading up to failure? How/why exactly are the V_p measurements connected (or disconnected) to AE localization? In other words,

the Vp measurements show evidence of a precursory signature but the AE data do not. Why is this?

And what is the micromechanical explanation for these observations?

L345: The relative drops in Vp as a function of pore-pressure seem to be minor; that is it looks like the reduction in Vp is roughly the same, irrespective of the pore-fluid pressure. What do the numbers mean next the diamonds in Figure 7B? The same can be said about duration of the velocity

reduction; the duration of the Vp reduction seems highest between the dry and Pp = 0.5 case and is roughly independent of pore-pressure when Pp > 5 MPa. Can the authors comment on what they

think might be contributing to the non-linear behavior and invariance to relative changes in pore-pressure?

L388: Should mention that the second prerequisite for unstable sliding is that material exhibits rate-weakening behavior.

L394: See also work by Affinito et al., 2023.

L397-401: I'm confused by the statement about precursory signals and slow events. Do you mean that the events in the datasets here also contain slow slip events?

L413-417: Do you mean the average "patch size" that slipped during the AE? And is this measured in a standard seismology way via corner frequency? Please explain. If I understand correctly, you mean that macroscopic roughness is smaller than the area that slips in a given AE, and thus, the fault roughness does not act as a barrier to slow down and/or impede the rupture? So presumably you get bigger AEs on smoother surfaces. But if this is true then there must be some microscopic barrier along the fault plane that limits the overall size of the AE such that an individual AE does not represent the failure of the entire fault plane?

L419-423: See also work by Cattania and Segall, 2021.

L430-431: How can both higher normal stress and higher fluid pressures both create a homogenous fault state?

L436: And also the lack of robust eq detection algorithms; for example the work by Ross et al., 2019 and Trugman and Ross, 2019 was facilitated by simply enhancing the existing eq catalog with template matching.

L442: Do you mean large stress changes throughout the laboratory seismic cycle?

L448: typo: is a result of

L447: Could be useful to have a plot that shows this. Something like precursor duration or nucleation period (however one defines this) plotted as a function of stress.

L451: This general idea about decreasing foreshock activity with normal stress is also seen in Marty et al., 2023 and is consistent with some observational studies of foreshocks (e.g.,

Abercrombie and Mori, 1996; Peng and Mori, 2022). This is also consistent with nucleation theory and the concept of a critical nucleation length scale H^* ; as H^* scales inversely with normal stress and if foreshocks track the nucleation stage (i.e., creeping area that makes up the nucleation zone)

then maybe one would expect a decrease in foreshock activity as H^* reduces.

Reviewer #1 (Remarks to the Author):

I have reviewed the paper entitled “Why are foreshock rare in nature? A laboratory perspective” by Goebel et al., as a potential manuscript to be published in Nature Communication. The authors report on laboratory experiments where they investigate the effect of increasing fluid pressure on the stick slip behavior of granite samples. They used three different sample configurations: 1) dry saw-cut smooth surfaces; 2) in situ fractured samples and 3) the same as in 2 but thermally cracked to understand the interaction of changing stiffness and hydraulic properties on the mechanical behavior. To inform the mechanical data they use an array of PZTs to retrieve the ultrasonic wave velocity (active source) and passive Acoustic Emission (AE). From the experiments and AE data emerge a very complex interaction between the main fault zone and the surrounding rocks as a function of fluid pressure that is also highlighted by the AE.

I find the idea behind the paper very interesting, however, due to the large amount of data I got lost in understanding what is the real focus of the authors. As the title reads I would have expected a different type of paper. Here the authors mainly focus on the effect of fluid pressure on fault slip behavior so that I think the title is way too general since not all faults are overpressurized.

Response: We thank the reviewer for his comments that helped clarify several aspects of the original publication. We added more specificity to the title to guide the reader’s expectation right from the beginning. The main point of the paper is that many previous lab-tests have been too simplistic and the trade-offs between different boundary conditions (e.g., fluid pressure, roughness, stress, confining pressure, damage) are important to explain the obvious discrepancy between foreshocks in lab and nature.

In general the paper is well written, even though I found many typos throughout the manuscript. I think that the authors should work on reorganize and clarify their data selecting what is really important for the message they want to send. Most of the time more data can only confuse the reader if there is not a clear explanation of the meaning of those data. In fact at the end of the manuscript I found myself confused on what data really supported their conclusions. Nonetheless, the data are very interesting and the methodology sounds good. In conclusion I think that the paper deserves publication however after some very major revisions and reorganization of the manuscript. I will list a series of comments below that I hope will help in improving the manuscript.

Sincerely
Marco Scuderi

Response: We would again like to thank the reviewer for his insightful comments. We revised the manuscript substantially to improve clarity of figure design and to comprehensively demonstrate the importance of each of the presented datasets. We

appreciate the reviewer's deferring opinion on how much data and how many experiments should be included and emphasize that we prefer a more wholistic view on preparatory processes instead of oversimplifying the observations by preselection, which may have been an issue that contributed to the core topic of this paper - i.e., the fundamental disconnect between lab observations and natural seismicity.

We improved the structure of the result section and more clearly separated the four main topics, i.e., 1) AE source types and micro-mechanics, 2) fault micro-structures, 3) pore volume increase before slip and 4) precursory signals as a function of Pp and roughness. We also made sure that all figures in part 4 have a similar layout.

In the manuscript, we wrote:

“Here, we show how different fault conditions, namely geometric complexity (i.e., fault roughness and damage) and pore fluid pressure affect precursory signals before failure. We demonstrate that moment tensor components become more dominated by earthquake-like double-couple mechanisms for rough faults at high pore pressures. Elevated pore fluid pressures and reduced roughness lead to more rapid acceleration of premonitory slip and shorter precursory periods. This process is likely governed by more homogeneous stress distributions and lower geometric complexity on both smooth and high fluid pressure faults, which can be quantified by overall focal mechanism variability.”

Line-by-line comments:

L34: why use Leeman as a reference in this context?

Citation removed.

L34-36: I do not find myself in agreement with this statement since the failure of intact rocks involve mechanical processes that are quite different from frictional sliding along pre-existing faults with gouge. Indeed, here it arises a problem that I found throughout the paper and I invite the authors to clarify. In their opinion is faulting (i.e. reactivation of a preexisting plane of weakness) a fracture of frictional processes? I read in the text jumping from one to another and this does not help the clarity of the paper.

We appreciate the reviewer's comment about the long-lasting discussion on the micromechanics of faulting which, in the lab, involves both fracture and frictional processes – perhaps at different space/time scales (see e.g., Bayart et al., Kwiatek et al., McLaskey et al., Chen et al.). Resolving this discussion is not a central focus of this paper. We changed the sentence to: “Precursory signals have long been recorded prior to intact rock fracture and the brittle failure of large asperities.”

Bayart, E., Svetlizky, I., & Fineberg, J. (2016). Fracture mechanics determine the lengths of interface ruptures that mediate frictional motion. *Nature Physics*, 12(2), 166–170.

<https://doi.org/10.1038/nphys3539>

McLaskey, G. C., & Glaser, S. D. (2011). Micromechanics of asperity rupture during laboratory stick slip experiments. *Geophys. Res. Lett.*, 38. <https://doi.org/10.1029/2011GL047507>

Kwiatek, G., Goebel, T. H. W., & Dresen, G. (2014). Seismic moment tensor and b value variations over successive seismic cycles in laboratory stick-slip experiments. *Geophysical Research Letters*, 41(16). <https://doi.org/10.1002/2014GL060159>

Chen, X., Carpenter, B. M., & Reches, Z. (2020). Asperity Failure Control of Stick–Slip Along Brittle Faults. *Pure and Applied Geophysics*, 177(7), 3225–3242.

<https://doi.org/10.1007/s00024-020-02434-y>

Goebel, T. H. W., Becker, T. W., Schorlemmer, D., Stanchits, S., Sammis, C., Rybacki, E., & Dresen, G. (2012). Identifying fault heterogeneity through mapping spatial anomalies in acoustic emission statistics. *Journal of Geophysical Research: Solid Earth*, 117(3), 1–18.

<https://doi.org/10.1029/2011JB008763>

L45: crack alignment, can the authors be clearer on what they mean with this process? Is this coalescence on microcracks in an intact rock?

Changed to:

“crack alignment with respect to each other” as an expression of an overall homogenization of crack orientations.

L59-61: stress corrosion processes are quite different from subcritical crack growth. Can the author be more specific?

Changed to: “Stress corrosion which is commonly thought to facilitate subcritical crack growth is governed by the availability of fluids and may control AE-foreshocks in some lab-tests (Das and Scholz, 1981; Main et al., 1992).”

Line 82: on the fault structure there are more references to add.

There certainly are but here and in many other places, we are limited but the journal format. Changed to: “(e.g., Goebel et al., 2013b; Scuderi et al. 2017)”

L84: here the authors talk about frictional break-down processes that are different from fracture propagation as invoked before. Can the authors clarify which constitutive framework they want to use? There is nothing wrong to treat both (fracture and friction) but the authors should be clear about it.

This sentence simply highlights similarities between lab and nature. We refer the reviewer to the above citations that demonstrated the difficulty of separating fracture and frictional processes especially during the early stages of rupture nucleation.

L106: here the author mention a confining pressure of 120MPa but I do not see experiments at this pressure from table S1

This pressure refers to the first experiment in Table S1.

109-111: can the author mention that the natural fracture was induced by notch on the sample. I found this information buried in the methods but I think it is important to mention here to help the reader understand the comparison with saw-cut.

Changed to: "eight samples were partially cut, then fractured at $P_c=75\sim\text{MPa}$ and finally reloaded after increasing the confinement to $P_c=150\sim\text{MPa}$."

L121-122: "... lower bulk moduli and higher across-fault crack densities for thermally treated samples". I do not understand this sentence. Is higher for increasing?

Changed to: "The resulting microcrack damage caused strongly non-linear stress-strain relationships at low confining pressures. Loading at higher confining pressures resulted in more-linear stress-strain relationships but with relatively lower bulk moduli for thermally-treated samples "

127-128: about the surface smoothing. The authors here refer to figure 2 where naturally fractured rock are reported. However, also based on the microstructures, I have some doubts in the interpretation of surface smoothing since the authors show the creation of gouge that should at this point control fault mechanics and the characteristics of stick-slip. Why the authors highlight surface smoothing and not gouge creation associated with strain localization?

Changed to: "This strength reduction is caused by the combined effects of lower effective stresses, surface smoothing and gouge production with cumulative slip"

Figure 2: Coupling this figure with table S1 I see that there are no experiments at high fluid pressure and "normal" granite samples. I see only the experiment 24 that arrives at the fluid pressure reported in fig 2c. However this is thermally damaged but I do not see specified anywhere in the caption. Can the authors clarify? Also, why do they compare experiments with different boundary conditions? As mentioned in the text the thermal cracking changes the elastic moduli of the rock so that stick slip behavior should be different from the "normal" intact rock.

We appreciate this comment and agree that other processes may contribute. However, our AE results demonstrate the primary importance of pore pressure changes for all samples at $P_c=150\text{ MPa}$. In the spirit of the reviewer's initial comments, we thus focus on the primary controlling parameter for shear-type events that is resolved in our experiments which is pore pressure (see also Fig. 3).

L130-131: this statement should be clarified and contextualized. Clearly the presence of fluids influence the microscale deformation via chemical reaction at the grain scale, however, I am confused on the reasoning of why high fluid pressure should increase small scale deformation. Indeed, since fluid pressure is isotropic in all direction and lower the effective stress in my opinion it should decrease stress at the contact scale possibly reducing the real contact area and doing so inhibit shear deformation.

This is an interesting observation. Here, we refer to a purely observational result based on the observed AE moment tensors, which demonstrate that pore pressure increases the relative proportion of shear type events and shear deformation. We clarify this statement: "Higher fluid pressures also promote relatively more shear deformation expressed in dominant shear-type AE events at the micro-scale"

L134-138: I find these sentences a little confusing since it is not clear to me if the authors are referring to intact rock deformation or gouge. The authors should be more clear about it and maybe use a different terminology. Pore space collapse is usually referred to intact rock deformation after Yielding at P^* (i.e. Zhang et al., 1990). In the context of gouge deformation cataclasis and grain comminution are processes that control porosity. However, when talking about gouge deformation differences should be made between high strain and low strain domain since porosity will be heterogeneous.

We agree and clarify the sentence: "Intact rock compaction due to highly compressive stress fields in dry samples is accommodated at the microscale by seismically detectable pore space collapse."

Figure 3: the authors use this figure to show that shear type events are dominant. I would be curious to know where those events come from, if they are located in the damage zone or within the sample and so controlling shear deformation and stick slip. In the methods I read that the AE location uncertainty is between 0.5 to 3mm and from the microstructures in figure 4 I see that the gouge is about 100 micron thick. At line 139 to 156 the authors seem to relate these AE to the microscale of gouge deformation. How can they be sure that the AE are representative of microscale deformation at the contact scale of the gouge? I would also be curious of the central frequency of the piezoelectric sensor they use. Is that high enough to resolve such detail?

Gouge thickness varies between 0.1 to 1 mm. Our AE results robustly resolve locations within the gouge zone which also accommodates most strain. (see e.g., Dresen et al. 2020; Goebel et al. 2014). The argument for dominant AEs within the gouge is simple, because the AE distribution peaks at the center of the fault and decays as a power-law. The resonance frequency of the sensor is 2 Mhz.

Dresen, G., Kwiatak, G., Goebel, T., & Ben-Zion, Y. (2020). Seismic and Aseismic Preparatory Processes Before Large Stick-Slip Failure. Pure and Applied Geophysics, 177(12), 5741–5760.

<https://doi.org/10.1007/s00024-020-02605-x>

Goebel, T. H. W., Becker, T. W., Sammis, C. G., Dresen, G., & Schorlemmer, D. (2014). Off-fault damage and acoustic emission distributions during the evolution of structurally complex faults over series of stick-slip events. Geophysical Journal International, 197(3), 1705–1718.

<https://doi.org/10.1093/gji/ggu074>

L160-163: I think that this information should go much earlier in the paper since it is a great importance. In fact if 2 out of three experiment show stable behavior, why the authors show only the experiment that show instability? How much those instability are producible?

We added a sentence at the beginning of the Result section: "Axial loading at high confinement led to stick-slip in six tests and stable sliding in two experiments as discussed in the following section." We do not exclude any experiment (or recorded data) from the analysis, i.e., both stick-slip and stable sliding.

L166-167: to me it is very counterintuitive the fact that fluid pressure stiffen the rock. In fact, some measurements we made in our lab on sandstones (unpublished) shows the opposite. Looking at the paper the authors refer to, the stiffening seems related to crack density. If this is true does it mean that the sample that show stick slip has much less crack density? The authors report figure S4 as a measurement, but since they also have P-wave velocity, do they see a stiffening from such measurements?

We cannot comment on unpublished data but Fig. S4 demonstrates the stiffening effect of elevated pore pressure which seems very plausible. Replacing compressible air in the rock pores and cracks with incompressible fluid leads to a stiffness increase that is roughly proportional to the applied pore pressure (see Fig. S4, right panel).

We see a more rapid increase in seismic velocity during initial axial loading of saturated and high pore pressure samples. Nevertheless, interpreting seismic velocity changes in terms of stiffness/elastic properties is challenging since V_p and V_s depend differently on fluid content (Budiansky & O'Connell, 1976), which is why we prefer the more direct measurement in Fig. S4.

L228-234: I would like to give another perspective on these data. Since fault porosity is quite low, looking at the microstructure, I think that the pre-seismic fluid volume increase can be due to the opening of interconnected fractures in the damage. Indeed, high damaged fault is the one that show this signal. If such dilation was happening in the gouge, high and low damage should have shown the same amount of dilation since the gouges look very similar in porosity.

Certainly, these are interesting comments, which we had already discussed in the following paragraph.

L237-245: I agree with this interpretation.

L250-251: However, it seems that increasing fluid pressure also decrease the duration of the

entire seismic cycle. I think that to make a direct comparison the authors should express the duration of pre-seismic creep as a percentage of recurrence interval. The different recurrence time it means that fluid pressure also affect frictional healing that can generate a different premonitory stage.

We added in the text: “Both stable sliding and stick-slip exhibit extended periods of slow, premonitory slip which starts before pore volume dilation. However, increasing pore pressures significantly shorten the duration of slow slip before failure (Fig. 6). The respective relative time periods expressed as percentage of recurrence intervals where 39, 23, 24, and 17 % for rough faults at 0, 0.5, 5 and 15 MPa pore pressure and 4 % for the dry, smooth faults.” Note that we provide both relative and absolute time frames to better describe the actual lab times scales and their relative differences.

L307-308: To have the definitive proof if this is true the authors should compare rough dry faults at the same mean stress as the high fluid pressure. Could it be due to a reduction of effective stress?

The smoothing of the underlying stress field is observed in AE moment tensors and minimum pair rotation angles as explained in this paragraph. We added: “Relatively smoother stress fields are measured by using the minimum rotation angle between pairs of moment tensors” see also Goebel et al., 2017

Goebel, T. H. W., Kwiatek, G., Becker, T. W., Brodsky, E. E., & Dresen, G. (2017). What allows seismic events to grow big?: Insights from b-value and fault roughness analysis in laboratory stick-slip experiments. *Geology*, 45(9), 815–818. <https://doi.org/10.1130/G39147.1>

L347-348: I think this is a little bit an overstatement in particular for the micro-structural characteristics.

We removed the first part of this sentence.

L376-382: Can the author clarify this statement. In my opinion, larger contact area should inhibit accelerated creep and unstable behavior. If we think to the combination of adhesion friction theory and the slip weakening law, in order to have a stress drop and slip acceleration the contact area must decrease as a function of slip. Indeed, many experimental works can be found on phyllosilicate rich faults where seismic slip is suppressed by the alignment of phyllosilicate foliae that increase the contact area, in some cases the contact area saturates and is expressed by a zero value of the rate parameter b in the context of rate and state friction.

We completely agree with the reviewer and would like to point out that equation (2) and the respective paragraph exactly suggest an inverse relationship between real area of contact, A (which depends on grain size, d (see L 355: $A = f(d^m/n)$) and sliding velocity where $V = f(d_0/d)^m$). On L 376 – 378, we wrote: “The above constitutive relation predicts that fault slip velocity is strongly affected by the real area of contact, A, with larger values of A hindering more rapid slip propagation.”, which matches the reviewer’s intuition. We further clarify this paragraph: “This area also evolves with subsequent stick-slip due to wear production, grain comminution and alignment of phyllosilicates. Similarly, shorter preparatory phase and rapid

slip acceleration on smooth faults are explained by smaller contacts and more rapidly changing real area of contact compared to rough faults with larger d."

L393-395: However, elevated fluid pressure can also change the critical rheologic stiffness $k_c = (b-a)s_n/D_c$ by decreasing the effective stress and push the fault to stability. In my opinion this is a more plausible mechanism.

We added: "... pore pressures increase stability through increasing fault damage zone stiffeness, and reducing effective normal stress and frictional parameters."

L396-401: The information that aseismic creep has premonitory signals is the first time that it is mentioned. Also slow slip events are mentioned here for the first time. Can the author rework this entire section since I do not understand what is the focus and the message they want to send.

We removed this particular paragraph to provide a more focused discussion.

L424-425: Can the authors motivate this statement? Fluid pressure is an isotropic pressure that should act equally in all directions. So why it should act as a smoothing factor? What the author say it might be possible but under near lithostatic condition, while, from a rapid calculation here the authors are under sub hydrostatic condition (i.e. $\lambda \sim 0.2$).

The fluid pressure effect is observed in AE moment tensors and minimum pair rotation angles (see above comment). We believe that the difference originates from relative load distributions across rock-air vs. rock-fluid-filled cracks. We added: "A potential cause for these pore pressure effects is the relative increase in geometric and stress field homogeneity when loads are distributed across asperities and fluid-filled pore space compared to air-filled pores."

L429-431: This sentence is quite confusing because there is at least a contradiction, how can normal load increase if also fluid pressure is increasing?

We corrected this sentence: "We hypothesize that fault contact stresses progressively evolve to a more homogeneous state if either normal load, fault slip, or pore pressure increase, leading to compressed foreshock activity."

L445-446: I agree regarding the confining pressure, however, how can the author say it about the fluid pressure?

We removed pore pressure from this sentence.

Methods:

L491: I think that more specification on how the fracturing was performed should be given here.

We added: "The natural fractures were guided by two 25~mm notches on opposite sides of the samples, inclined at 30 degree to the loading."

L493-495: where can I see these data?

Stick-slip and stable sliding is shown in Fig. S3. Since RMS roughness estimates is a standard engineering measurement, we do not show the results but the full spectrum of asperity heigh distributions can be seen in an early publication which we added here.

*Goebel, T. H. W., Kwiatek, G., Becker, T. W., Brodsky, E. E., & Dresen, G. (2017). What allows seismic events to grow big?: Insights from b-value and fault roughness analysis in laboratory stick-slip experiments. *Geology*, 45(9), 815–818. <https://doi.org/10.1130/G39147.1>*

L497-498: can the authors clarify this sentence? Is there more than one patch?

We clarified this sentence: "All fluid-saturated tests were conducted at constant pressure boundary conditions facilitated through a central borehole, to simulate the effect of a fault patch in nature, embedded within a hydraulically connected fault damage zone. "

Reviewer #2 (Remarks to the Author):

Summary

This work shows why foreshocks are relatively rare in nature than the observations in the laboratory. They provide laboratory experiments that show the richness of laboratory foreshock is strongly dependent on fault pressure. They investigate the multiple factors that might influence foreshock – roughness, pre-existing damage, and pressure. They found even a small magnitude of pressure can significantly reduce the preparatory phase of lab earthquakes. The paper further investigates the microstructural behavior of fault zones. The authors claim that the existence of pore pressure might be the reason for the relative absence of foreshock in natural fault systems.

Review Summary

I agree that foreshocks are more abundantly observed in the lab than actual earthquakes. As the authors discussed, I thought the paucity of foreshocks might have originated from the detection threshold. But now I am convinced that, as this paper says, the rock-fluid interaction might also influence the preparatory phase of natural earthquakes. The experiments seem to be fairly well-designed, and the analyses are convincing. I recommend publishing this paper.

Minor Comment

Table S1 shows the experimental setups for each 10 experiments conducted. However, it is hard to recognize which experiment is shown in the figures except figure 3. For example, figure 2 shows three experimental results but does not show which experiment they are. It would be helpful if authors could put an ID for each experiment shown in plots wherever possible, as done in figure 3.

We appreciate this recommendation and now added experiment IDs to Figs. 2 and 5.

Reviewer #3:

Summary:

Goebel et al. report on a series of laboratory experiments designed to understand nucleation processes leading up to laboratory stick-slip instabilities. The authors test the influence of fault roughness, damage, and pore-fluid pressure on nucleation processes, foreshock activity, and elastic wave properties. They focus most of their analysis on the role of roughness and pore-fluid pressure. One of the main takeaways from the work is that fluid pressure, roughness, and damage all act to modulate nucleation processes and potentially precursory signals. Specifically, enhanced stresses and pore-pressures seem to reduce precursory signals which may explain the lack of such signals in the Earth. Overall, I find this topic and set of experiments very interesting. I believe this work is suitable for publication once the authors address the following comments/questions below.

Major points/comments:

1. The authors show that increasing pore-fluid pressure decreases AE localization and reduces pre-seismic fault creep preceding failure. The authors also show that V_p decreases prior to failure with and without the presence of fluids. The authors claim that the duration and the overall magnitude over which V_p drops decreases with increasing pore-fluid pressure. I have a hard time observing these trends in the data in Figure 7. Based on the plots it appears that V_p decreases prior to failure, irrespective of pore-pressure. The magnitude and duration of the drop in V_p do change for pore-fluid pressures between 0-5 MPa, but for pressures > 5 MPa they are independent of pore-pressure magnitude (Figure 7B). I think this point is worth clarifying and acknowledging in the text since one of the major claims of this work is that increasing pore-fluid pressure reduces precursory signals.

Response: We thank the reviewer for his comments which we used to improve and clarify several aspects in the paper. To address the pore pressure comment, we rewrote the corresponding paragraph in the paper:

“In our experiments, the reduction in seismic velocities starts between 1,500 to 3,200 seconds before failure, which is substantially earlier than the measured onset of sample dilation and slow fault slip (Fig. 7B). The drop in seismic velocity is most pronounced and starts earliest for the dry, rough-fault experiment. Fluid-saturated tests exhibit shorter periods of velocity reduction before failure with little difference between tests at 5 and 15 MPa. Shorter periods of precursory P-wave velocity reduction are in approximate agreement with more rapid fault activation at elevated pore pressure.

Changes in seismic velocity are caused by a combination of effects related to crack closure and new crack formation with increasing deviatoric stresses. In addition, the degree of fluid-saturation of the evolving damage and gouge zones affect velocity changes before failure (O’Connell and Budiansky, 1974).”

2. I found it difficult to keep track of the varying boundary conditions (e.g., changes in damage, pore-pressure, and roughness). It could help the flow of things if these were separated more clearly in the text and figures. For example, right now Figure 2 discusses the role of pore-fluid pressure and then Figure 5 transitions to fault damage state and then there is a transition back to pore-pressure with Figure 6. And the effect of roughness is embedded in there as well.

We improved the structure of the result section and more clearly separated the two main components, i.e., 1) microstructure and macro slip and 2) precursory signals before slip. We also made sure that all figures in the latter part have a similar layout more clearly showing results from tests with three different initial/boundary conditions: 1) smooth, dry fault, 2) rough dry fault, 3) rough fluid saturated fault. In the manuscript, we added the following paragraph at the end of the Introduction:

“Here we examine how different fault conditions, namely fault roughness, damage and pore fluid pressure affect precursory signals before failure. The following section is divided into four parts: 1) We first discuss differences in acoustic emission source types and 2) associated fault structural differences in microscopy images. 3) We then show differences in pore volume change before slip in samples with low and high microcrack damage. Lastly, 4) we examine the effect of pore pressure and roughness on preparatory signals, i.e., accelerating fault slip, seismic velocity reduction, AE event localization, AE rate increase and focal mechanism variability. Our results suggest that elevated pore fluid pressures at seismogenic depth likely reduce the extent of preparatory phases before slip. Underlying rapidly accelerating nucleation processes at large depth may be difficult to detect outside of high-resolution lab studies.”

3. The effect of damage on precursory acoustic signals is not discussed. Were AE and Vp data collected for these experiments? It seems like this would be a useful thing to discuss (if there are indeed acoustic data that go along with these experiments) because they play an important role in the main conclusion of the

paper. Specifically, the effects of damage and the role it plays in generating (or reducing) precursory AE and V_p signals could be elaborated on in the discussion.

Response: We agree with the reviewer and now specifically discuss the effect of damage on AEs and pore volume changes. Seismic velocities are essentially identical for these tests.

“Preparatory signals are amplified by the degree of pre-existing micro-crack damage; however, this effect depends on confining pressure. Low confining pressure up to 75~MPa leads to pronounced non-linear stress-strain relations and notable pore volume dilation before in-tact rock fracture. At higher confining pressures, stress-strain relations become increasingly more linear even for high-damage samples but with respective lower elastic moduli. AE event source types on high-damage samples are comprised of up to 9 % more shear type events, are slightly less localized and exhibited higher rate peaks during slip when loaded at the same pore pressure as low damage samples. These observations suggest that the degree of fault damage may play an important role in amplifying precursory signals, however more tests at high pore pressure and low effective pressure are needed to verify this observation.”

Comments:

L34: Not sure if Leeman et al., 2016 should be cited here. Normal stress dependence of b-value was shown in the work of Riviere et al., 2018 and also differential stress in Scholz, 2015. Yamashita et al., 2021 also looked at surface roughness on 1-meter faults.

We added Riviere et al., 2018 and Yamashita. Scholz, 2015 does not investigate precursory signals.

L36: Could also cite Scholz, 1968.

Okay, added

L45: Also strongly influenced by fault unlocking and slip rate. L47: Can you clarify what you mean by short-term signals?

“Short-term signals stem from processes associated with rupture nucleation and the commencement of detectable fault slip”.

L56: I think this should be Johnson et al., 2017; Hulbert et al., 2019 also showed that various aspects of slow and fast stick-slip could be predicted using AE data.

Change to: “(e.g. Hulbert et al., 2019; Johnson et al., 2013; Karimpouli et al., 2023; Rouet-Leduc et al., 2017; Shreedharan et al., 2021b).

L57: Can the authors comment on how they might think this model of crack propagation and coalescence would extend to cases where there foreshocks and AEs are generated from microscale processes occurring in a finite layer of fault gouge.

The extension to the asperity / crack model is explained in detail in the Discussion section (Eq. 2, L360ff). We added: “This framework can be extended to the non-linear behavior of interacting frictional contacts (see Discussion and Barbot, (2019)).”

L78: See also Affinito et al., 2023; Samuelson et al., 2008; Proctor et al., 2020.

Added citations.

L115: What is the roughness of the smoothed surfaces compared to the surfaces that are fractured in-situ?

This is discussed in the Method Section and Goebel et al., 2017

L124: Can you clarify what you mean by micro and macro scales?

Changed to: Changes in pores pressures affect slip mechanics at the scale of sub-millimeter AE events and macro slip events.

L136: Do the data in Figure 2 correspond to the “rough” faults that were fractured in-situ or the smooth faults that were surface polished? Please clarify.

*These are rough faults, see caption: “Higher fluid pressure lead to more shear type events and fewer compaction events during stick-slip sliding on **rough faults.**”*

L139: Comparing Figure 2B and 2C one can tell that the number of shear type events has decreased with increasing fluid pressure. However, this trend is more challenging to

observe when comparing the pore-pressure increases within the same experiment in Fig 2C. The same can be seen in regards to the tensile type events. Can the authors comment on why they think this trend is masked in Figure 2C? Is this somehow linked to differences in fault roughness? Is the effective normal stress held constant between the three experiments shown in figure 2 or does it change accordingly with pore-pressure?

We do not see evidence for the trend being 'masked' in Fig. 2C. Note how the relative number of tensile events decreases, and shear events increase at $P_p=35$ MPa. The relative increase is from ~45% at 5 MPa to more than 60% shear type events at 35 MPa. (relative contributions always add up to 100% in this plot)

L163: This is consistent with rate-state theory and previous lab experiments from Leeman et al., 2016; Scuderi et al., 2016;2017.

We agree with reviewer, see Discussion: "Pore fluid pressures reduce the effective stress and cause a moderate increase in damage zone stiffness – which both promote stable sliding on limited size fault patches in the lab (Leeman et al., 2016; Scuderi et al., 2016). [...] pore pressures increase stability through increasing fault damage zone stiffness and reducing effective normal stress, and frictional parameters (Scuderi and Collettini, 2016).

L166: The critical stiffness, K_c , will also change with effective normal stress as pointed out above. Also possible that D_c , and $a-b$ change with normal stress (Scuderi et al., 2016).

See response to previous comment.

Figure 5B- Are the high and low damage experiments conducted at the same pore-pressure? For the low damage case, why is the fluid vol. decreasing? Is the sample compacting during this phase?

Pore pressures are now also listed in the caption. We added "Pore volume reduction for low damage sample is due to axial compression."

L251/Figure 6: Why do the slip displacement curves start near the end of the seismic cycle as opposed to the beginning of the seismic cycle? And are the stress drops the

same for all events shown in Figure 6? Or does stress drop decrease with increasing pore-pressure? Is it possible that variations in roughness are playing a role in the pre-seismic changes in pore-pressure? Or do the events in Figure 6 correspond to the same shear strain/cumulative offset? It would be useful to plot the pre-seismic change in fluid volume as a function of pre-seismic slip. Presumably you would find that these scale systematically with one another? Also, I suppose that part of the reason you have more pre-slip with decreasing pore-pressure is simply because the recurrence interval of the seismic cycles are longer from higher effective stresses and more frictional healing.

We appreciate this compilation of thoughts, and a lot of these comments were addressed in the previous manuscript. Fig. 6 shows displacements and velocities relative to the time of failure (see x-axis label). We added the start of premonitory slip as relative percentage of seismic cycle duration (see above response) which further demonstrates the difference in slip acceleration to failure (see legend in Fig. 6b) and shows that differences in effective stress alone do not explain the observation.

As suspected by the reviewer both roughness and pore fluid pressure fundamentally influence preparatory behavior before macroscopic slip events (see L 256ff). Note that the onset of pore volume increase and premonitory slip are not aligned (L 249: Both stable sliding and stick-slip exhibit extended periods of slow, **premonitory slip which starts before pore volume dilation.**), and cannot easily be correlated. We suspect that pore volume changes are driven by both new damage generation and gouge zone dilation (L 237ff: Two observations suggest that not only gouge layer dilation but also crack opening across the fault damage zone contributes to the measured change in pore volume).

L261: Shouldn't the onset of inelastic creep be determined from a stress vs strain plot as opposed to stress vs time? Also, it seems that the onset on inelastic creep is not very sensitive to changes in pore-pressure. The location of the stars are ~ in the same location on all three stress curves. It would be useful to extend the slip displacement curves back in time so that they encompass this deviation from linear-non-linear behavior.

Part of the confusion here is that unlike in direct shear and gouge experiments, we get significant deformation within both the fault core, the surrounding damage zone and the host rock which affects stress accumulation and subsequent fault activation. The onset of non-linear stress is not equal to the onset of detectable fault slip. To

avoid confusion, we changed the sentence: “[.] differential stresses exhibit onsets of non-linear behavior that is progressively closer to failure with increasing pore pressures, implying that stress accumulation rates are more sensitive to pore pressure changes than peak stress.[...] Non-linear stresses before failure are expected to be a result of gradual fault activation, slip and microcrack damage evolution.”

L271: See also Shreeharan et al., 2020;2021. Scuderi et al, 2016

Citations added.

L287: what are the uncertainties in the AE locations? Please clarify.

Method: “[...] travel time residuals of less than 0.5 μ s, thereby minimizing the likelihood of erroneous detection and locations. AE location uncertainty was between 0.5 and 3 mm”

L294-296: This is similar to what Bolton et al., 2023 observed for slow vs fast slip stick-slip events. The AE localization effect preceding failure is also seen the experiments from Marty et al., 2023.

These observations were already discussed in the Introduction, L41-44, but we added the suggested citation.

L338: The AE localization does indeed seem to disappear with the addition of fluids. However it's interesting that V_p decreases prior to failure, irrespective of the pore-fluid pressure. Do the authors believe this drop in V_p is related to shear/damage localization leading up to failure? How/why exactly are the V_p measurements connected (or disconnected) to AE localization? In other words, the V_p measurements show evidence of a precursory signature but the AE data do not. Why is this? And what is the micromechanical explanation for these observations?

P-wave velocity depends on the ratio of bulk moduli over density and may not be directly correlated with AE localization. Rather we find that P-wave velocity is more directly related to AE rates and damage accumulation (Fig. 9a) but pore

space geometry, orientation and degree of fluid saturation are also important (see e.g., Pandey et al., 2023).

Pandey, K., Taira, T., Dresen, G., & Goebel, T. H. (2023). Inferring damage state and evolution with increasing stress using direct and coda wave velocity measurements in faulted and intact granite samples. *Geophysical Journal International*, 235(3), 2846–2861. <https://doi.org/10.1093/gji/ggad390>

L345: The relative drops in V_p as a function of pore-pressure seem to be minor; that is it looks like the reduction in V_p is roughly the same, irrespective of the pore-fluid pressure. What do the numbers mean next the diamonds in Figure 7B? The same can be said about duration of the velocity reduction; the duration of the V_p reduction seems highest between the dry and $P_p = 0.5$ case and is roughly independent of pore-pressure when $P_p > 5$ MPa. Can the authors comment on what they think might be contributing to the non-linear behavior and invariance to relative changes in pore- pressure?

We agree that, overall, the differences between dry and fluid saturated experiments are larger than differences within fluid saturated tests at different pore pressures. This is observed for fault slip, P-wave velocity, source mechanisms, and slip velocity. Nevertheless, the differences between pore pressure tests exhibit the same systematics for each of the presented measurements. Note that both duration of velocity decrease before failure and percentage are systematically reduced with higher pore pressures, except for experiment 23 – and we also acknowledge that the velocity reduction is much more modest than differences in duration:

“Evolving fault structure and damage produce pronounced drops in seismic P-wave velocities before failure, with modest differences between the experiments (Fig. 7). High pore pressure tests show the least amount of velocity reduction whereas dry and low pressure tests show more significant decrease in seismic velocity before failure.”. Numbers in B have been removed.

L388: Should mention that the second prerequisite for unstable sliding is that material exhibits rate-weakening behavior.

Good point. We added: "A condition for stick-slip within this framework is rate-weakening friction (i.e., $b > a$)."

L394: See also work by Affinito et al., 2023.

Citation added.

L397-401: I'm confused by the statement about precursory signals and slow events. Do you mean that the events in the datasets here also contain slow slip events?

Paragraph has been removed.

L413-417: Do you mean the average "patch size" that slipped during the AE? And is this measured in a standard seismology way via corner frequency? Please explain. If I understand correctly, you mean that macroscopic roughness is smaller than the area that slips in a given AE, and thus, the fault roughness does not act as a barrier to slow down and/or impede the rupture? So presumably you get bigger AEs on smoother surfaces. But if this is true then there must be some microscopic barrier along the fault plane that limits the overall size of the AE such that an individual AE does not represent the failure of the entire fault plane?

Yes source dimensions are determined in a standard way and AEs on average tend to get bigger on smooth surfaces. Source dimension was determined from seismic moment and corner wavelength which is now added to the description. The barrier hypothesis is indeed stated here: "large-scale heterogeneity and separation length between asperities provide efficient barriers that hinder rupture propagation." Note that roughness wavelength distributions provide not a deterministic length scale of seismic sources but more of a probabilistic rupture arrest scale.

L419-423: See also work by Cattania and Segall, 2021.

Added.

L430-431: How can both higher normal stress and higher fluid pressures both create a homogenous fault state?

Sentence was corrected, see previous comment.

L436: And also the lack of robust eq detection algorithms; for example the work by Ross et al., 2019 and Trugman and Ross, 2019 was facilitated by simply enhancing the existing eq catalog with template matching.

We agree that improved data processing helps but the point here is that (space/time) scale of nucleation processes may entirely be out of reach because of the required proximity of instruments to the source region.

L442: Do you mean large stress changes throughout the laboratory seismic cycle?

L448: typo: is a result of

Changed to: "Large stress drops during slip in the lab due to different boundary conditions may significantly exceed those in nature"

L447: Could be useful to have a plot that shows this. Something like precursor duration or nucleation period (however one defines this) plotted as a function of stress.

Unfortunately, our experimental data is not sufficient to systematically plot duration as a function of stress.

L451: This general idea about decreasing foreshock activity with normal stress is also seen in Marty et al., 2023 and is consistent with some observational studies of foreshocks (e.g., Abercrombie and Mori, 1996; Peng and Mori, 2022). This is also consistent with nucleation theory and the concept of a critical nucleation length scale H^* ; as H^* scales inversely with normal stress and if foreshocks track the nucleation stage (i.e., creeping area that makes up the nucleation zone) then maybe one would expect a decrease in foreshock activity as H^* reduces.

We agree and add: "Lower roughness and increased normal stress are expected to reduce the nucleation length of unstable ruptures leading to rapid slip acceleration and short foreshock periods. (Marty et al., 2023)."

REVIEWERS' COMMENTS

Reviewer #1 (Remarks to the Author):

I have reviewed the re-submission of the paper by Goabel et al. I believe that the authors made substantial changes to the manuscript that greatly improved the clarity of the results and the interpretation. At this stage I believe that the paper is suitable for publication without further changes. I list below two very minor changes:

Line 9: I would rephrase as "we examine the effects of geometric and stress heterogeneity on preparatory processes. Using acoustic emissions on samples with different roughness, damage and pore pressure we aim at unveil pre-seismic signals"

L197: I would change the title to be "Comparison of fault micro-structural features" or something a little more descriptive.

Marco Scuderi

Reviewer #3 (Remarks to the Author):

I thank the authors for thoroughly addressing all of my comments and questions raised in the previous round of revisions. The quality and clarity of the manuscript has greatly improved and I recommend the manuscript be published. I have a few minor comments for the authors to think about/consider adding to the discussion.

1. L510-511 it seems like it would be worth pointing out that even in the laboratory the changes in wavespeed prior to failure are very small ($< 1\%$) which could easily become masked without high resolution measurements.

2. Based on the work presented here and other lab studies, is it fair to say that most of the precursors that have been observed and documented in the laboratory occur only under simplistic conditions (e.g., dry, rough fault zones)? So maybe precursors are not "robust" characteristics of lab data but are simply seen more frequently because experiments are typically conducted under simplistic conditions that facilitate precursory activity? Also, what about the possibility that the laboratory precursors are often observed on small scale faults (\sim cm-size) that slip more or less simultaneously during failure (i.e. they behave like 1D spring slider model) which is not completely analogous to what happens on a real fault (i.e., where only a small segment of larger fault zone fails). Is it possible that the discrepancy in scale is also leading to an observational bias?

Reviewer #1 (Remarks to the Author):

I have reviewed the re-submission of the paper by Goabel et al. I believe that the authors made substantial changes to the manuscript that greatly improved the clarity of the results and the interpretation. At this stage I believe that the paper is suitable for publication without further changes. I list below two very minor changes:

Line 9: I would rephrase as "we examine the effects of geometric and stress heterogeneity on preparatory processes. Using acoustic emissions on samples with different roughness, damage

and pore pressure we aim at unveil pre-seismic signals"

We changed this sentence to: "Here, we examine the effects of geometric and stress heterogeneity on premonitory slip and associated acoustic emissions on faults with different roughness, damage and pore pressures."

L197: I would change the title to be "Comparison of fault micro-structural features" or something a little more descriptive.

Done

Marco Scuderi

Reviewer #3

I thank the authors for thoroughly addressing all of my comments and questions raised in the previous round of revisions. The quality and clarity of the manuscript has greatly improved and I recommend the manuscript be published. I have a few minor comments for the authors to think about/consider adding to the discussion.

1. L510-511 it seems like it would be worth pointing out that even in the laboratory the changes in wave speed prior to failure are very small ($< 1\%$) which could easily become masked without high resolution measurements.

We added: "Seismic velocities can vary by as much as 10% during the fracture of intact rocks and only 0.2 to 1.2% prior to stick-slip in faulted samples.

2. Based on the work presented here and other lab studies, is it fair to say that most of the precursors that have been observed and documented in the laboratory occur only under simplistic conditions (e.g., dry, rough fault zones)? So maybe precursors are not "robust" characteristics of lab data but are simply seen more frequently because experiments are typically conducted under simplistic conditions that facilitate precursory activity?

The current observations and the documented effect of roughness, heterogeneity and pore pressure do indeed support this inference.

Also, what about the possibility that the laboratory precursors are often observed on small scale faults (\sim cm-size) that slip more or less simultaneously during failure (i.e. they behave like 1D spring slider model) which is not completely analogous to what happens on a real fault (i.e., where only a small segment of larger fault zone fails). Is it possible that the discrepancy in scale is also leading to an observational bias?

We agree that scale is important, especially as it relates to roughness and slip nucleation. In the Discussion, we quantify the effect of roughness-scales on foreshock development, which could also be done in nature if respective measurements are available: L372ff:

“The effect of surface roughness on the development of foreshock sequences may depend on the relative spatial scales of respective seismic events and average separation-distance between load-bearing asperities.”